# Topological signatures in regulatory network enable phenotypic heterogeneity in small cell lung cancer

Lakshya Chauhan[1,2†], Uday Ram[1,2†], Kishore Hari[1], Mohit Kumar Jolly[1]*

[1]Centre for BioSystems Science and Engineering, Indian Institute of Science, Bangalore, India; [2]Undergraduate Programme, Indian Institute of Science, Bangalore, India

**Abstract** Phenotypic (non-genetic) heterogeneity has significant implications for the development and evolution of organs, organisms, and populations. Recent observations in multiple cancers have unraveled the role of phenotypic heterogeneity in driving metastasis and therapy recalcitrance. However, the origins of such phenotypic heterogeneity are poorly understood in most cancers. Here, we investigate a regulatory network underlying phenotypic heterogeneity in small cell lung cancer, a devastating disease with no molecular targeted therapy. Discrete and continuous dynamical simulations of this network reveal its multistable behavior that can explain co-existence of four experimentally observed phenotypes. Analysis of the network topology uncovers that multistability emerges from two teams of players that mutually inhibit each other, but members of a team activate one another, forming a 'toggle switch' between the two teams. Deciphering these topological signatures in cancer-related regulatory networks can unravel their 'latent' design principles and offer a rational approach to characterize phenotypic heterogeneity in a tumor.

*For correspondence:
mkjolly@iisc.ac.in

[†]These authors contributed equally to this work

## Introduction

'Genotype controls phenotype' has been a prevalent paradigm across multiple biological contexts (*Orgogozo et al., 2015*). However, past few decades have revealed in many biological organisms that a fraction of cells in a genetically identical population can behave differently from others, even under nearly identical environmental conditions. This 'phenotypic heterogeneity' usually refers to 'non-genetic' variations among individual cells in a genetically homogeneous scenario (*Grote et al., 2015*). In microbial populations, this heterogeneity can manifest as variation in morphologies, growth dynamics, metabolic signatures, and response to antibiotics. It can enable 'bet-hedging', thereby providing cell populations or organisms with higher fitness especially in fluctuating environments (*Ackermann, 2015*). Another advantage of phenotypic heterogeneity is division of labor and the possibility of cooperation among phenotypically distinct subpopulations (*Armbruster et al., 2019*). Phenotypic heterogeneity is an evolvable trait and can, in turn, shape evolutionary trajectories at genomic levels too, as seen in bacteria, yeast (*Bódi et al., 2017*; *van Boxtel et al., 2017*), and more recently in cancer cell populations (*Salgia and Kulkarni, 2018*). Therefore, decoding mechanistic underpinnings of emergence of phenotypic heterogeneity remains crucial.

Two generic mechanisms proposed for phenotypic heterogeneity are stochastic gene expression and multistability. Stochastic fluctuations in levels of various biomolecules can trigger the cells to reversibly switch their phenotypes, with important functional implications in tolerance to antibiotics (*Balaban et al., 2004*), in enabling the latency of HIV (*Weinberger et al., 2005*), and in long-term resistance to anticancer drugs (*Inde and Dixon, 2018*; *Ramirez et al., 2016*; *Sharma et al., 2010*), highlighting common principles of population behavior across living systems (*Ben-Jacob et al.,*

2012; Jolly et al., 2018). Cancer has been largely thought of as a genetic disease driven by accumulated mutations in key oncogenes and tumor suppressor genes involved in hallmarks of cancer such as increased proliferation and decreased cell death (Hanahan and Weinberg, 2011). Thus, evolution in cancer has been mostly postulated as a Darwinian process of clonal selection (Wooten and Quaranta, 2017), therefore promoting gathering of resource-heavy data through international consortiums such as The Cancer Genome Atlas (TCGA) or International Genome Cancer Consortium (ICGC) primarily at a genomic level. However, increasing evidence has shown that a cancer cell's phenotype is not solely driven by its genotype (Brock et al., 2009; Hu et al., 2020; Sharma et al., 2019) and that non-genetic mechanisms such as stochastic gene expression and/or multistability can propel metastasis (Jolly and Celià-Terrassa, 2019; Lee et al., 2014) and resistance to various therapies (Meyer et al., 2020; Miura et al., 2018; Mohanty et al., 2020; Pisco et al., 2013; Shaffer et al., 2017; Spencer et al., 2009; Su et al., 2019) – the two major unsolved clinical challenges in cancer. Consistently, no unique mutational signature has been yet identified for metastasis; instead, phenotypic adaptability is considered to be a hallmark of metastasis (Celià-Terrassa and Kang, 2016; Welch and Hurst, 2019). Therefore, it is essential to investigate the emergent dynamics of regulatory networks that can give rise to multistability and consequently accelerate tumor aggressiveness.

Here, we elucidate the dynamics of a regulatory network underlying phenotypic heterogeneity in small cell lung cancer (SCLC) – a highly metastatic cancer that accounts for 15% of all lung cancer cases, has a 7% five year survival rate, has no molecular targeted therapy, and shows rapid relapse to a treatment-refractory phase after initial response to chemotherapy and radiotherapy that remains to be the standard of care for SCLC for over half a century. Therefore, SCLC has been placed in the category of recalcitrant cancers (Gazdar et al., 2017; Udyavar et al., 2017). Nearly all SCLC cases show genomic inactivation of TP53 and RB1 (George et al., 2015); thus, phenotypic heterogeneity observed in SCLC cannot be explained based on the mutational status (Rudin et al., 2019). Here, we demonstrate that phenotypic heterogeneity in SCLC can be explained by simulating the dynamics of a regulatory network underlying SCLC using both continuous and discrete modeling approaches. The four predominant phenotypes that this network enables can be mapped on to recently identified molecular subtypes of SCLC – ASCL1$^{high}$/NEUROD1$^{low}$, ASCL1$^{low}$/NEUROD1$^{high}$, ASCL1$^{high}$/NEUROD1$^{high}$, and ASCL1$^{low}$/NEUROD1$^{low}$. Further analysis of SCLC network topology reveals that multistability in this complex network of 33 nodes and 357 edges emerges due to two teams of players in the network that are operating against one another – players of both teams tend to activate and be activated by members of the same team, while they inhibit and are inhibited by members of the other team, thus forming an effective 'toggle switch' between the two teams. Intriguingly, this topological signature trait is specific to this SCLC regulatory network; perturbing the network disrupts this signature and leads to disappearance of these four phenotypes. Furthermore, data analysis shows that ASCL1$^{low}$/NEUROD1$^{low}$ subtype can be sub-classified into ASCL1$^{low}$/NEUROD1$^{low}$/YAP1$^{low}$/POU2F3$^{high}$ and ASCL1$^{low}$/NEUROD1$^{low}$/YAP1$^{high}$/POU2F3$^{low}$ classes. Together, our results unravel the latent design principles of the complex SCLC regulatory network and offer a rational framework to decode the phenotypic heterogeneity in SCLC through the lens of network topology.

## Results

### Discrete and parameter-agnostic continuous simulations of the SCLC network result in similar phenotypic distributions

We simulated the emergent dynamics of a SCLC master regulatory network constructed from gene expression signatures, which contained 33 nodes and 357 edges (Udyavar et al., 2017; Figure 1A). To obtain the steady-state (i.e. phenotypic) distributions corresponding to this complex network, we implemented two complementary approaches – one of them is a discrete parameter-independent Boolean modeling approach using Ising model formalism and an asynchronous update mode (Font-Clos et al., 2018), and the other is RACIPE (Random Circuit Perturbation) (Huang et al., 2017), a parameter-agnostic approach that uses a set of coupled ordinary differential equations (ODEs) with parameters sampled over a wide biologically relevant range.

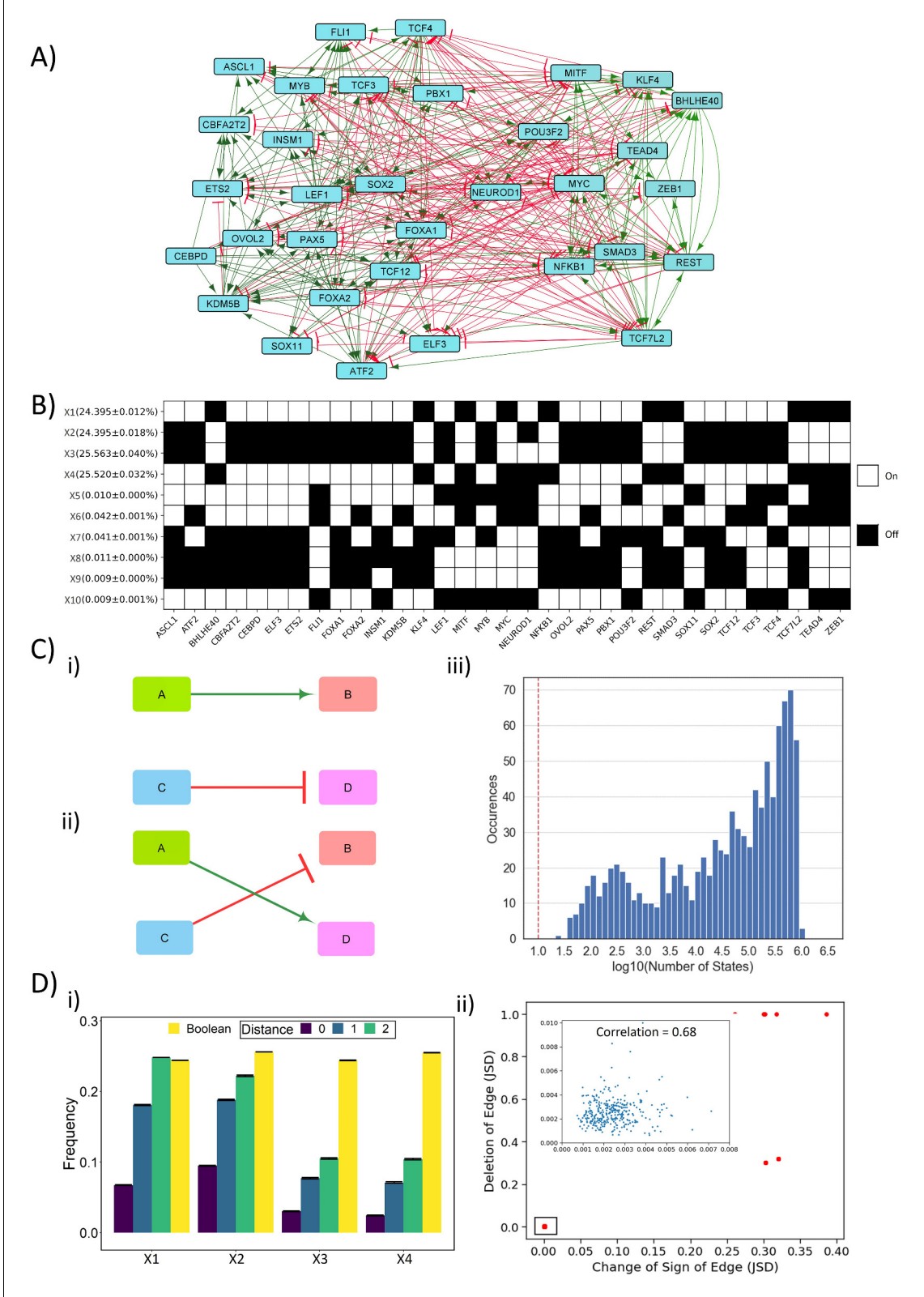

**Figure 1.** Dynamic simulations of SCLC network. (A) SCLC regulatory network, in which green nodes denote activation and red nodes denote inhibitory interaction. (B) Steady states achieved from asynchronous Boolean update, using Ising model and $2^{20}$ initial conditions (for results corresponding to $2^{25}$ initial conditions, see *Supplementary file 1a*). Each row is a steady state, and each column is a node in the network. Dark cells represent node 'off' (0) and blank ones represent node 'on' (1). The frequency of each state is reported to the left of each row in percentage as mean ±

*Figure 1 continued on next page*

*Figure 1 continued*

standard deviation over three replicates. (C) (i–ii) Schematic representing edge swapping strategy for network randomization, where (ii) is a randomized network for the 'wild-type' (WT) network (i). (iii) Distribution of number of steady states for 1000 randomized networks corresponding to SCLC WT network. Red line shows the number of states obtained for the SCLC WT network. (D) (i) Comparison of steady-state frequencies obtained via RACIPE and Boolean. The frequency of four dominant states obtained in Boolean (X1–X4) is shown in yellow bars. The other three bars show the frequency of RACIPE states identical to corresponding Boolean states (distance = 0) and the cumulative frequency of states which are less than or equal to n nodes having different values (node value = 0 in RACIPE and 1 in Boolean or vice versa) (distance = 1, 2). Results over three replicates are reported as mean ± standard deviation (error bars). (ii) Correlation plot between Jensen–Shannon divergence (JSD) for the case of edge deletion and JSD for the case of reverting the sign of the corresponding edge. Each dot denotes an edge that has been perturbed. Inset shows zoomed-in view for the highlighted small box. The mean and standard deviation of Pearson's coefficient correlation values for this scatter plot, over three replicates, is 0.851 ± 0.003.

In the Ising model formalism, each state is represented as a Boolean vector of N elements, where N (=33 here) is the number of nodes in the networks. Based on randomly chosen set of initial conditions (n = $2^{20}$ here), multiple simulations capture the ensemble of steady states obtained. The discrete-time asynchronous network dynamics considered here is simulated via

$$s_i(t+1) = \begin{cases} +1, & \sum_j M_{ij}s_j(t) > 0 \\ -1, & \sum_j M_{ij}s_j(t) < 0 \\ s_i(t), & \sum_j M_{ij}s_j(t) = 0 \end{cases}$$

where $s_i(t)$ denotes the expression levels of node $i$ at time $t$. $s_i$ = +1 means that the node is highly expressed (i.e. 'ON' state), otherwise $s_i$ = −1. M depicts the interaction matrix of the network. $M_{ij}$ = 1 indicates that node $i$ promotes the expression of node $j$, $M_{ij}$ = −1 implies that node $i$ inhibits the expression of node $j$, $M_{ij}$ = 0 implies no regulatory interaction from node $i$ to node $j$.

For this network, we obtained 10 unique steady states, among which four of them (X1–X4) had a frequency of 24.3–25.5% each, while the remaining six states (X5–X10) had frequency less than 0.1% (*Figure 1B*). To characterize these steady states further, we calculate the frustration of each state, which, using the analogy of asymmetric spin glass on a graph, is defined as the fraction of network edges that are frustrated in that state. An edge from node $i$ to node $j$ is said to be frustrated in the state $\{s_i\}$ if $M_{ij}s_is_j < 0$, i.e. if the values of node $i$ and node $j$ do not follow the set of regulatory interactions between them. As expected, states with high frequency (~24.3–25.5% each) had low frustration (0.14) and those with lower frequency (<0.1% each) have higher frustration (0.37) (*Supplementary file 1a*, *Tripathi et al., 2020b*).

To understand the role of network topology of the SCLC network in enabling these steady states, we perturbed the network topology by picking up random pairs of edges and swapping them (*Figure 1C,i,ii*). This process is repeated for many iterations to ensure that the resultant network (referred to as the random network(s) henceforth), is very different from the original SCLC network (referred to as the 'wild-type' [WT] network henceforth). We created 1000 such random networks and simulated them using the above-mentioned Ising model formalism. Importantly, a majority of these networks gave rise to a much larger number (i.e. $10^4$–$10^6$) of steady states as compared to the WT network (*Figure 1C,iii*). Furthermore, none of these steady-state distributions obtained from random networks had any overlap with the steady-state distribution of WT network, suggesting that the steady-state distribution of WT network is unique to its topology.

To gain further confidence in the role of network topology in enabling the steady-state distribution of WT network, we used RACIPE that generates an ensemble of kinetic models (set of coupled ODEs) corresponding to a network topology, each with a different set of parameters sampled uniformly from a wide but biologically relevant parameter space. Each kinetic model is simulated using an ODE solver to obtain different steady states obtainable from different initial conditions. We generated $10^6$ models and simulated each of them with 1000 initial conditions, from which we obtained the ensemble of steady states and discretized them (see Materials and methods) to obtain a distribution that can be compared with the output obtained from a Boolean modeling framework. RACIPE enables a much larger number of states as compared to Boolean, which is not surprising given its continuous nature. The net frequency of the X1–X4 states obtained via RACIPE is 22%. However, all of the top 20 states obtained by RACIPE were close enough to one or more of the top four Boolean

states (X1–X4); there were a maximum of 2 of 33 nodes whose values were different (0 in Boolean state and 1 in RACIPE state or vice versa). These top 20 states contribute to a total of 54% frequency of all steady-states identified by RACIPE (*Supplementary file 1b*). This minor difference observed in state compositions prompted us to a more quantitative comparison. We calculated the net frequency of states that have only one or two node values different than that seen in corresponding Boolean state (X1–X4). The total frequency of RACIPE state, together with its 'close-enough' states, was similar to that of corresponding Boolean states (*Figure 1D,i*), thus endorsing a consistency in gene expression programs identified by Boolean and RACIPE models. This similarity between RACIPE and Boolean outputs reinforces that network topology has a crucial role in enabling the resultant steady-state distributions of the SCLC network.

Next, we investigated the resilience of the WT SCLC network by generating two types of 'mutant' networks. Each 'mutant' network has an edge either deleted or its sign reversed (from activation to inhibition and vice versa). Thus, we had $2*357 = 714$ such 'mutant' or 'perturbed' networks for which we calculated the steady-state distributions using the Ising model ($2^{20}$ initial conditions). To quantify the degree of similarity between the steady-state distributions of WT SCLC network and that of a 'perturbed' network, we used an information-theory metric known as Jensen–Shannon divergence (JSD), which ranges from 0 to 1 (*Lin, 1991*). JSD = 0 indicates identical distributions, and JSD = 1 indicates completely non-overlapping ones. As a control, JSD = 1 was obtained for all 1000 random networks with respect to the WT network, given that there is no overlap of states. Intriguingly, we observed that most single-edge mutations (deletions or sign reversal) had a negligible effect on dynamics of the WT SCLC network (JSD < 0.01 for 678 of 714 perturbed networks). Also, the JSD values of mutated networks with either change correlated well (*Figure 1D,ii*), suggesting a consistent contribution of the edge mutated in enabling steady-state distributions of the SCLC network. Importantly, among the 36 edge perturbations that led to a higher JSD, i.e. disturbed the frequency distribution, 24 of them were incoming edges for NEUROD1 (*Supplementary file 1c*). This observation suggests that NEUROD1 can play a key role in maintaining the robustness of SCLC network dynamics and subsequently in enabling phenotypic heterogeneity. Put together, these results revealed that the SCLC network is quite resilient to single-edge perturbations and suggests the possibility of multiple layers of redundancy or reinforcement in the SCLC network to enable this specific steady-state distribution.

## Similarity between simulation and SCLC experimental data based on node correlations

To investigate the degree of resilience or reinforcement within the SCLC network, we performed pairwise correlation between all 33 nodes in the network. Strikingly, based on the top four states obtained from Boolean simulation of the WT network, we observed that 32 of 33 nodes formed two groups of nodes, such that within a group, the nodes were positively correlated, but across the groups, each pairwise correlation was negative (all nodes except NEUROD1; *Figure 2A,i*). Therefore, the network seemed to form two competing 'teams' or groups of players. Interestingly, one of the groups (top triangle in *Figure 2A,i*; called as group A henceforth) contained ASCL1, INSM1, FOXA1, and FOXA2, all of which are implicated in neuro-endocrine phenotype(s) in SCLC. On the other hand, the other group (lower triangle in *Figure 2A,i*; called as group B henceforth) contained REST, SMAD3, and ZEB1 are suggested markers/regulators of non-neuroendocrine (NE) or mesenchymal subtype(s) of SCLC (*Borromeo et al., 2016*; *Tlemsani et al., 2020*).

We defined a metric J to quantify the cumulative strength of relationship among different nodes in the network, which quantifies the degree of similarity in the two 'teams' and mutual competition among them.

$$J = \sum_{i=1}^{i=22}\sum_{j=1}^{i<j} P_{ij} + \sum_{i=24}^{i=33}\sum_{j=24}^{i<j} P_{ij} - \sum_{i=24}^{i=33}\sum_{j=1}^{j=22} P_{ij}$$

Here, P denotes the correlation matrix, and $P_{ij}$ denotes Pearson's correlation coefficient between node $i$ and node $j$; the key to the indices is given in table shown in *Figure 2*. For correlation matrix obtained for Boolean simulations for the WT network, the coefficient values are either 1 or −1, and the value of J metric is calculated to be 496. However, for the 1000 random networks, the values of J are quite small; the average value of J across 1000 networks is 11 (*Figure 2A,ii*), thus suggesting

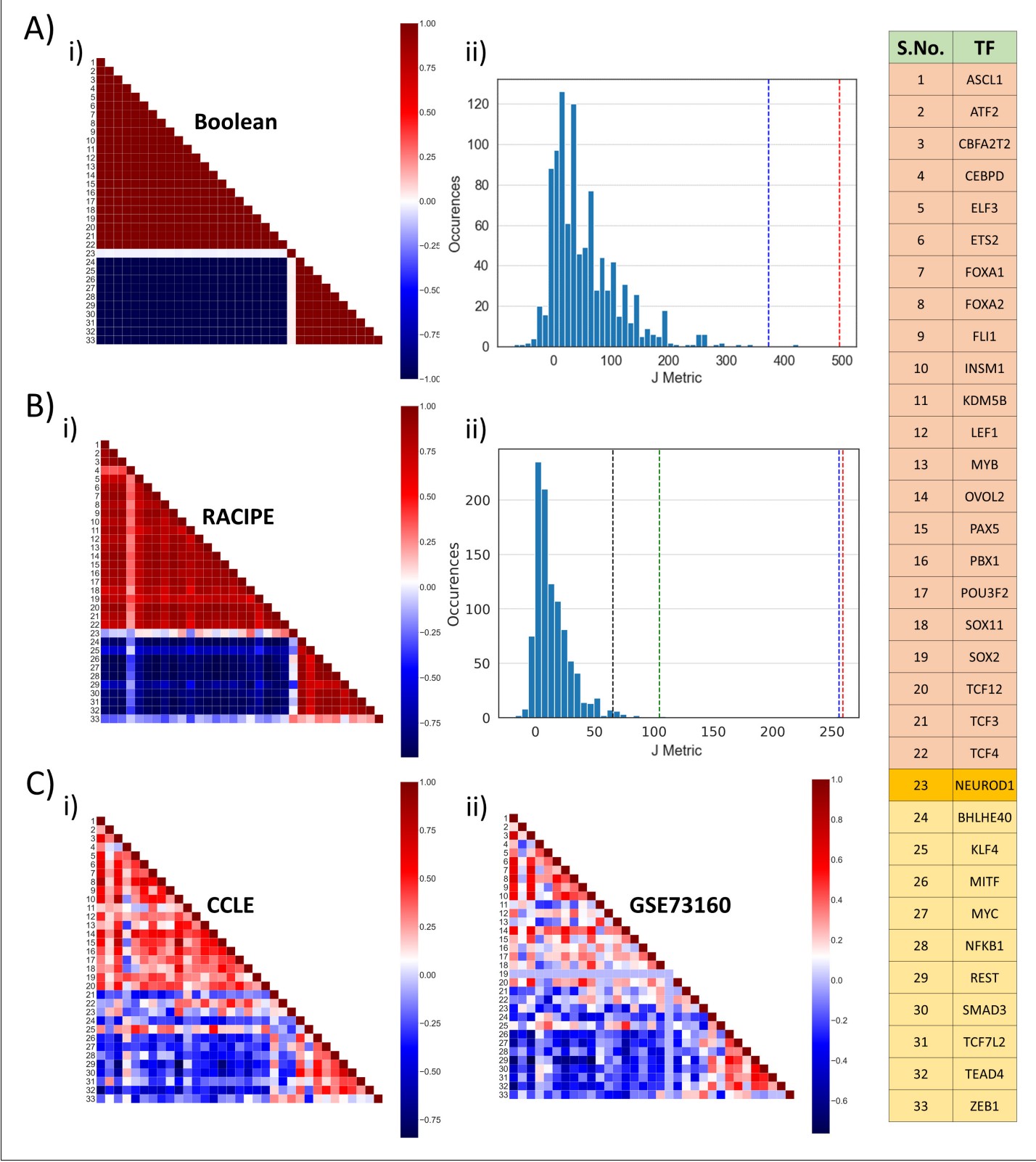

**Figure 2.** Identification of two 'teams'. (**A**) (i) Pearson's correlation matrix for Boolean simulations of WT SCLC network. Each node represents the correlation coefficient for pairwise correlations, as shown in adjacent colormap. (ii) Distribution of the values of J metric for 1000 random networks for Boolean simulations. Red dotted line shows the value of J (=496) for Boolean simulations (**A, i**); blue dotted line shows the same (J = 373.05) for RACIPE simulations (**B, i**). (**B**) (i) Same as (**A, i**) but for RACIPE simulations. (ii) Same as (**B, ii**) but for cases when Pearson's correlation coefficient values are

*Figure 2 continued on next page*

Figure 2 continued

sampled continuously from [−1,1]. Red dotted line shows the value of J for wild-type Boolean (**A**, i) and blue dotted line shows the value of J for Wild-type RACIPE (**B**, i). Black and green dotted lines show the value of J for CCLE (**C**, i) and GSE73160 (**C**, ii). (**C**) Same as (**A**, i) but for CCLE (i) and GSE73160 (ii). Details of indices 1–33 are available in the rightmost table.

that this metric can be a quantitative method to distinguish between WT and random networks and that this observed feature of two 'teams' of players is unique to the WT topology.

Similar trends of two 'teams' were seen in RACIPE simulation data (*Figure 2B,i*), albeit the value of J was lower (=373.05), largely because the values of Pearson's correlation coefficients for all (non-discretized) RACIPE pairwise comparisons will lie between −1 and +1 on a spectrum, instead of discretized values of −1 and +1 seen for Boolean simulations. For a fairer comparison, we recalculated the value of J metric for 1000 random networks now using continuous values of $P_{ij}$ (see Materials and methods). The mean value of J obtained from these 1000 networks was 14.16, a value much smaller than those corresponding to simulations of the WT network ($p<0.01$ for two-tailed z-test) (*Figure 2B,ii*). Furthermore, correlation matrices obtained from two experimental datasets – CCLE (n = 52), GSE 73160 (n = 63) – also implied the existence of these two 'teams', an observation that was strengthened by their value of J metric being significantly larger ($p<0.01$ for two-tailed z-test) than that of 1000 random networks considered here (*Figure 2B,ii*).

Put together, this excellent agreement among Boolean and RACIPE simulations and experimental data through correlation matrix underscores that the steady states obtained via simulations can be mapped on specific biological phenotypes seen in experimental data and endorses the idea about two 'teams' of molecular players that may inhibit each other to enable these steady states.

## SCLC network topological signatures underlie the emergence of biological phenotypes

The remarkable agreement among Boolean and RACIPE simulations, and the endorsement of the correlation patterns seen in simulations with the experimental data, leads us to the hypothesis that the underlying network topology is fundamental to the existence of these distinct phenotypes. To test this hypothesis, we formulated a metric to quantify the topological influence of one node on another, named as influence matrix (Inf). A path between two nodes (A and B) in a network of length l is defined as a series of connected edges starting at node A and ending at node B. While the interaction matrix (M) only quantifies the effect of nodes on each other for a path length of one, the influence matrix considers path lengths of up to $l_{max}$ to calculate this effect. Each element of the influence matrix, $Inf_{ij}$, is a number between −1 and 1, calculated as a weighted sum of net effect of all paths of length less than $l_{max}$ from ith node to jth node (see Materials and methods). $Inf_{ij} > 0$ indicates net activation, $Inf_{ij} < 0$ suggests net inhibition, and $Inf_{ij} = 0$ implies no net effect.

The influence matrix for path length $l_{max} = 10$ obtained for the WT network was very similar to the corresponding correlation matrix for RACIPE (*Figure 3A,i, Figure 3—figure supplement 1*) This similarity elucidates that the players in the two 'teams' are highly likely to effectively activate one another, but players belonging to different teams are likely to inhibit each other. To quantify this similarity, we calculated two metrics – R1 and R2 – bearing in mind that while the correlation matrix is symmetric, the influence matrix need not be. R1 is the correlation coefficient obtained from a scatter plot of regressing correlation coefficients between node i and j ($P_{ij}$) and corresponding influence matrix element ($Inf_{ij}$). R2 is defined the same as R1 but for $Inf_{ji}$ instead of $Inf_{ij}$. Both R1 and R2 values are high (>0.85) and statistically significant (*Figure 3A,ii*) and show a saturating trend for increasing path length (*Figure 3—figure supplement 2A*), thus we chose $l_{max} = 10$ for further analysis from influence matrix. The high values of R1 and R2, together with high coefficient of determination (*Figure 3A,ii*), reinforce that the influence matrix of WT SCLC network is quite similar to corresponding correlation matrix. Similar trends seen for correlation matrix from CCLE endorsed that the influence matrix can not only capture the patterns for steady states, but also can reflect biological phenotypes seen in SCLC (*Figure 3—figure supplement 2B*).

Next, we checked the specificity of the influence matrix to the WT network. We next regressed the elements of RACIPE correlation matrix of WT SCLC network against corresponding elements for influence matrix of one of the 1000 random networks we had simulated using asynchronous Boolean model formalism. Thus, we calculated 1000 values of R1 and R2 for all the random networks; none of

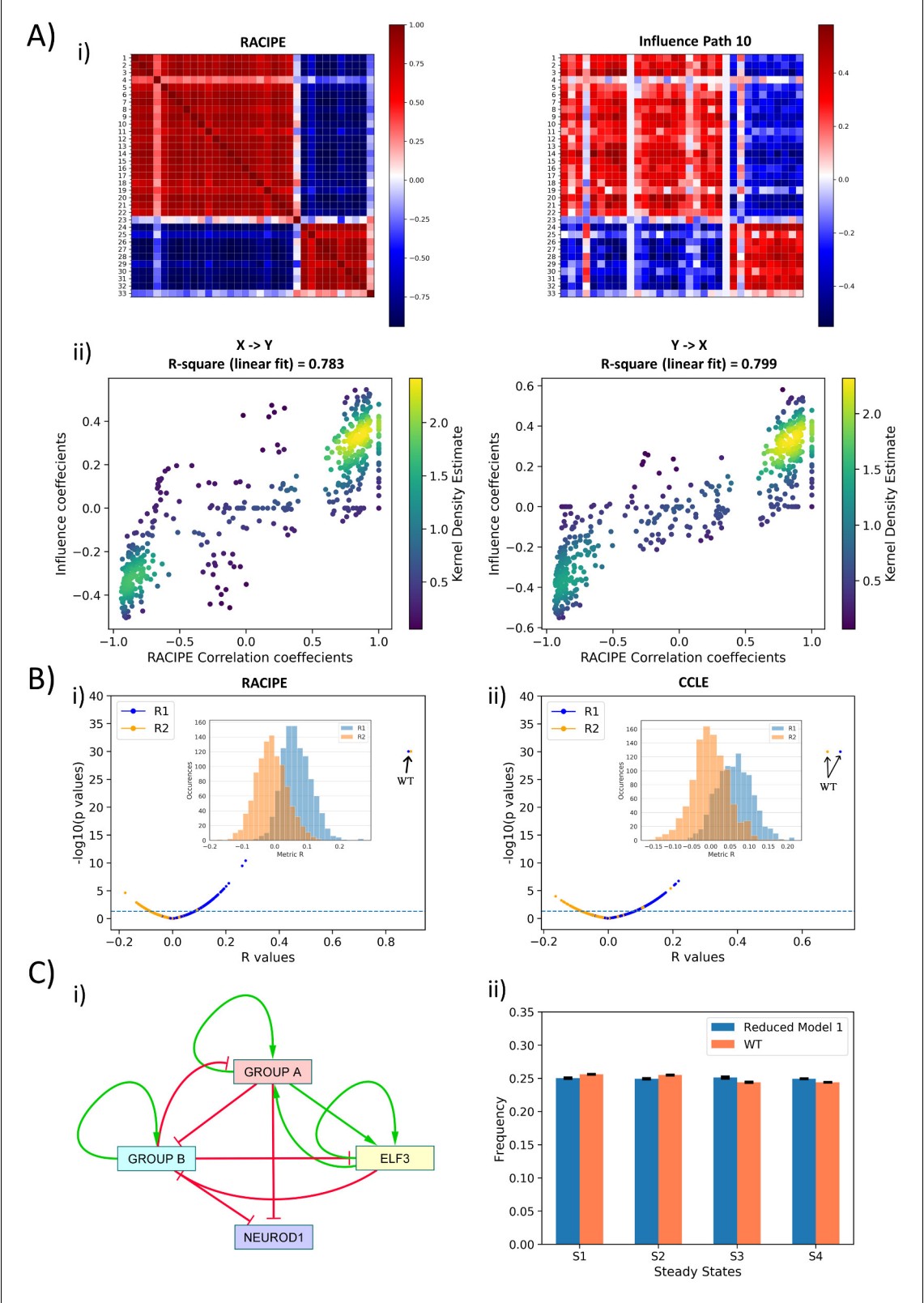

**Figure 3.** Topological features underlying the two 'teams' in SCLC network. (**A**) (i) (Left) Pearson's correlation matrix P for RACIPE simulations of WT SCLC network ($P_{ij} = P_{ji}$). (Right) Influence matrix Inf for WT SCLC network for path length = 10. Heatmap denotes $P_{ij}$ and $Inf_{ij}$ values, respectively. (ii) (left) Scatter plot for $P_{ij}$ and $Inf_{ij}$ values (for i < j, i.e. values in upper triangle of Inf and P matrices); (right) scatter plot for $P_{ji}$ and $Inf_{ji}$ values (for i > j, i.e., values in upper triangle of Inf and P matrices). R1 and R2 denote corresponding correlation coefficient values. Colorbar represents Kernel density

*Figure 3 continued on next page*

*Figure 3 continued*

estimate of points in the scatter plot. (**B**) (i) Volcano plot of all R1 and R2 metrics for RACIPE WT correlation matrix and random networks' influence matrices. Inset shows histogram of R1 and R2 values for corresponding cases. Horizontal line for p=0.05. Arrows point to R1 and R2 values of WT SCLC network. (ii) Same as (i) but for correlation matrix from CCLE. (**C**) (i) Reduced model derived from influence matrix; red bars show inhibition, and green arrows show activation. (ii) Steady-state frequencies obtained from reduced model one and that of WT SCLC network (n = 3).

The online version of this article includes the following figure supplement(s) for figure 3:

**Figure supplement 1.** Influence matrix based on Boolean simulations for states with various frustration levels Influence matrix in *Figure 3,A,i* is based on simulations from RACIPE.

**Figure supplement 2.** Topological features.

them were anywhere close to R1 and R2 of the WT SCLC network; many of these correlations were not even statistically significant (*Figure 3B,i*). Similar results were seen in the influence matrix from CCLE (*Figure 3B,ii*). Thus, we concluded that this influence matrix is unique to the WT SCLC network.

Put together, the influence matrix can be a more meaningful readout of interaction matrix in terms of effect of node *i* on node *j*. The influence matrix provides further credence to the idea that the SCLC network contains two 'teams' of players that are mutually inhibitory and self-activatory. To strengthen the concept of these two 'teams' of genes working together in a more qualitative sense, we constructed two reduced models from 'effective' edges calculated both from the interaction matrix and from the influence matrix (for path length = 10) (see Materials and methods) (*Figure 3C, ii*; *Figure 3—figure supplement 2C,i*). We performed asynchronous Boolean simulations for these reduced models using the Ising formalism. To compare the output of these reduced models with that of the WT SCLC network, we transformed the most frequent states of the WT SCLC network (shown in *Figure 1B*) into a representation using the same nodes as that included in the reduced networks. We observed that both reduced models – one generated from interaction matrix and another from influence matrix – resulted in the same four states as that of the WT (*Figure 3C,ii*, *Figure 3—figure supplement 2C,ii*). However, the network obtained using the influence matrix was closer to WT in terms of steady-state frequency distribution as compared to the one obtained using the interaction matrix. These results suggest that influence matrix is a better representation of network topology as compared to the interaction matrix.

Put together, we defined an influence matrix to decode the relative strengths of bidirectional regulatory interactions between every possible pair of nodes in the network. Analysis based on the influence matrix established the concept of two 'teams' of players that are effectively inhibiting each other and activating themselves, thereby forming an 'effective' self-activating toggle switch (SATS). Such SATS have been shown to be multistable and tend to underlie phenotypic plasticity and heterogeneity in multiple cell-fate decisions (*Guantes and Poyatos, 2008*; *Lu et al., 2013*; *Sahoo et al., 2020*; *Zhou and Huang, 2011*). This 'teaming up' can potentially explain (1) why single-edge perturbations in the WT SCLC network are rarely disruptive in terms of steady-state distributions (because as long as an 'effective' mutual inhibition between the two teams and 'effective' self-activation in the teams is maintained, the phenotypes are likely to be robust attractors) and (2) why despite such dense and complicated (33 nodes, 357 edges) network, we obtain only four steady states (because the 'latent' network topology is fundamentally simplistic).

## Classifying experimental SCLC data based on ASCL1 and NEUROD1

The first classification of SCLC was based on differences in cell morphology: a 'classic' type of cells that grow as spherical aggregates of floating cells and another relatively less prevalent 'variant' type which grew as tightly adherent monolayer or as loosely adherent aggregates. The 'classic' type had relatively higher levels of NE markers such as ASCL1 (*Gazdar et al., 1985*). Next, SCLCs were categorized into two categories based on relative levels of ASCL1 and NEUROD1, both of which are key developmental nodes for pulmonary NE cells (*Poirier et al., 2013*). Functional contributions of ASCL1 and/or NEUROD1 in SCLC have since been extensively explored (*Borromeo et al., 2016*; *Ikematsu et al., 2020*; *Osborne et al., 2013*). Further characterization of SCLC proposed three different phenotypes – ASCL1$^{high}$/NEUROD1$^{low}$, ASCL1$^{low}$/NEUROD1$^{high}$, and a 'double negative' ASCL1$^{low}$/NEUROD1$^{low}$ (*Poirier et al., 2015*). In addition, a 'double positive' ASCL1$^{high}$/NEUROD1$^{high}$ state of SCLC was recently identified (*Baine et al., 2020*; *Simpson et al., 2020*). In a recent

classification (*Rudin et al., 2019*), ASCL1$^{high}$/NEUROD1$^{low}$ was labeled as SCLC-A subtype, ASCL1-$^{low}$/NEUROD1$^{low}$ was categorized as non-NE (and subcategorized to SCLC-P and SCLC-Y subtypes), and ASCL1$^{low}$ /NEUROD1$^{high}$ was marked as SCLC-N (also referred to as NE-V1 *Wooten et al., 2019*; *Figure 4A,i*). Besides, the NE-V2 subtype expressed ASCL1 and had relatively higher levels of NEUROD1 as that seen in ASCL1$^{high}$/NEUROD1$^{low}$ subtype; thus, NE-V2 can be conjectured to be ASCL1$^{high}$ /NEUROD1$^{high}$.

Intriguingly, our simulations (*Figure 1A,ii*) are able to capture all these four phenotypes – these were the four most dominant steady states that Boolean and RACIPE results converged to. Thus, we were able to classify the Boolean steady states into four phenotypes – A+N+, A+N-, A-N+, and A-N- (where A represents ASCL1, N represents NEUROD1) (*Figure 4A,ii*). Hierarchical clustering of experimental data – CCLE (*Figure 4B,i*) and GSE73160 (*Figure 4—figure supplement 1A,i*) – using ASCL1 and NEUROD1 levels depict a clear segregation into four large groups at an early level. These four clusters were indeed A+N+, A+N-, A-N+, and A-N- (*Figure 4B,ii*, *Figure 4—figure supplement 1A*, ii). Alternate clustering methods such as K-means revealed K = 4 as the optimal number of clusters, based on the average Silhouette width analysis (*Figure 4C,i*, *Figure 4—figure supplement 1B,i*), thus strengthening the results obtained from Boolean and RACIPE analysis. These four clusters obtained by K-means also split into A+N+, A+N-, A-N+, and A-N- (*Figure 4C,ii*, *Figure 4—figure supplement 1B*,ii) reminiscent of observations that a classification system employing only ASCL1 and NEUROD1 may be sufficient to describe these four biologically meaningful phenotypes (*Borromeo et al., 2016*).

Based on the 'groups' we identified from influence matrix, ASCL1 was found to be a part of group A while NEUROD1 did not belong to any group (*Figure 3*). Therefore, we examined whether the expression levels of genes in group A were high in samples identified as A+N- and A+N+. Indeed, in CCLE and GSE73160, we observed that A+N- and A+N+ subgroups had high levels of group A and low levels of group B; NEUROD1 did not show any group-specific pattern (*Figure 4—figure supplement 2*). This observation endorses that states identified by mechanistic modeling are recapitulated in vitro.

Given that our choice of using ASCL1 and NEUROD1 to cluster SCLC cell lines was based on experimental data, we conducted an unbiased analysis of using any 2 of 33 possible nodes for characterization (i.e. $^{33}C_2$ = 528 combinations) in their ability to define four SCLC phenotypes. Only 140 such combinations yielded four as the optimal number of clustering, and the ASCL1-NEUROD1 pair showed up in the top 5 of the node pairs with maximal resolvability (measured via average Silhouette width) in defining these four phenotypes. Thus, the ASCL1 and NEUORD1 pair featured among the top 1% (in top 5 of 528 possibilities) of gene pairs in characterizing SCLC heterogeneity (*Figure 4—figure supplement 3*).

## Extended subtype classification of SCLC into five phenotypes based on POU2F3 and YAP1

Besides ASCL1 and NEUROD1, YAP1 and POU2F3 have been reported as important regulators of SCLC, particularly for the non-NE phenotype (*Baine et al., 2020*; *Huang et al., 2018*; *Ito et al., 2016*; *McColl et al., 2017*; *Song et al., 2020*). Thus, we investigated the role of these players in defining SCLC phenotypes. We performed hierarchical clustering over the four genes of interest (ASCL1, NEUROD1, YAP1, POU2F3) on the CCLE dataset (*Figure 5A,i*) and GSE73160 (*Figure 5—figure supplement 1A*, i). We observed five clusters, and when projected on the ASCL1-NEUROD1 axis, two of these five clusters were both present in the third quadrant (A-N-), suggesting that non-NE phenotype (ASCL1$^{low}$/NEUROD1$^{low}$) may be divided into two sub-clusters based on levels of YAP1 and POU2F3 (*Figure 5A,ii*, *Figure 5—figure supplement 1A,ii*).

To decipher the five clusters better, we first performed K-means clustering for K = 4 for CCLE using the levels of ASCL1, NEUROD1, YAP1, and POU2F3 and observed the four clusters: SCLC-A1 (A+; n = 28), SCLC-N (N+; n = 12) and two non-NE phenotypes: SCLC-Y (Y+; n = 8) and SCLC-P (P +; n = 4) (*Figure 5B,i*). When K-means clustering for K = 5 was performed on the same dataset using the four above-mentioned genes, SCLC-Y (Y+; n = 8) and SCLC-P (P+; n = 4) had identical composition as the corresponding clusters for K = 4 (*Supplementary file 2a*; *Figure 5B,ii*). Among 12 cell lines classified as SCLC-N (N+), 11 of them were still classified as SCLC-N with the exception of CORL279, which was classified as A+N+. Interestingly, the SCLC-A (A+; n = 28) cluster obtained for K = 4 broke into two sub-clusters: 14 cell lines were classified as 'NE-V2' (A+N+) and remaining 14

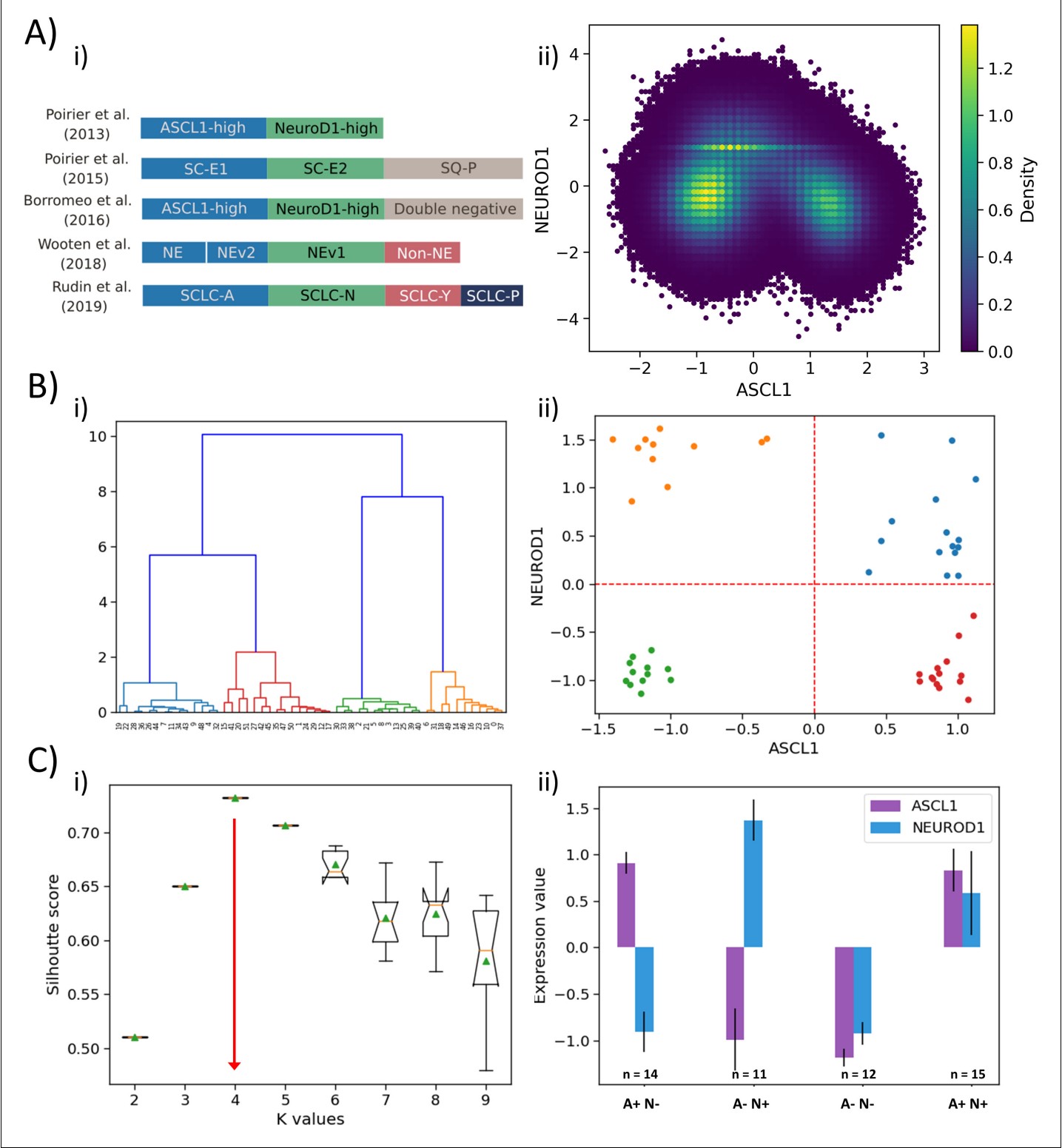

**Figure 4.** Classification of SCLC phenotypes based on ASCL1 and NEUROD1. (**A**) (i) Summary of classification of SCLC subtypes in the existing literature (adapted from *Rudin et al., 2019*). (ii) Density-based scatter plot for ASCL1 and NEUROD1 levels, as obtained via RACIPE modeling. (**B**) (i) Hierarchical clustering of CCLE samples using ASCL1 and NEUROD1. (ii) Scatter plot of normalized gene expression data for CCLE samples for NEUROD1 and ASCL1. Color coding in dendrogram and scatter plot are synonymous (i.e. refer to the same cluster). (**C**) (i) Average Silhouette width for different values of K for K-means clustering of CCLE samples for NEUROD1 and ASCL1. Error bars denote replicates of clustering attempts. (ii) Number of samples and ASCL1 and NEUROD1 expression values as seen in four clusters of CCLE samples obtained for K = 4.

*Figure 4 continued on next page*

*Figure 4 continued*

The online version of this article includes the following figure supplement(s) for figure 4:

**Figure supplement 1.** Analysis for GSE73160 using ASCL1 and NEUROD1.

**Figure supplement 2.** Gene expression of nodes involved in graph after clustering: Red bars indicate a node belonging to group A, and blue bar indicates belonging to group B.

**Figure supplement 3.** Clustering efficiency of other node pairs.

as SCLC-A. Similar trends were seen for hierarchical clustering, indicating the concurrence in assigning these different cell lines to SCLC phenotypes (*Supplementary file 2a*; *Figure 5B,ii*). This categorization suggests that the 'double positive' phenotype may be an intermediate one on the spectrum of ASCL1$^{high}$/NEUROD1$^{low}$ (SCLC-A) to ASCL1$^{low}$/NEUROD1$^{high}$ (SCLC-N) one, from the perspective of these four genes of interest. We compared the classification of the SCLC cell lines done using the levels of these four genes of interest with that done based on levels of a larger set of genes (*Wooten et al., 2019*) and obtained an overall good consistency, with a few (n = 5) exceptions where cell lines belonging to SCLC-A1 according to our clustering were found to belong to NEv2 as per the previous analysis (*Supplementary file 2a*). Performing the same procedure on another cohort of SCLC cell lines (GSE73160) – that showed consistent trends as for CCLE cohort (*Figure 5— figure supplement 1B*) – demonstrated that for K = 4 vs. K = 5, some SCLC-N cell lines were classified as the 'double positive' (*Supplementary file 2b*), endorsing that cells may switch to this hybrid phenotype from SCLC-A1 and/or SCLC-N, similar to observations in hybrid E/M phenotypes (*Tripathi et al., 2020a*).

Projecting the levels of these four genes of interest in SCLC cell lines on UMAP (Uniform Manifold Approximation and Projection) plots, we observed a neat splitting into five clusters – ASCL1$^{high}$/NEUROD1$^{low}$/YAP1$^{low}$/POU2F3$^{low}$, ASCL1$^{low}$/NEUROD1$^{high}$/YAP1$^{low}$/POU2F3$^{low}$, ASCL1$^{high}$/NEUROD1$^{high}$/YAP1$^{low}$/POU2F3$^{low}$ – and two subtypes of ASCL1$^{low}$/NEUROD1$^{low}$ – YAP1$^{high}$/POU2F3$^{low}$ and YAP1$^{low}$/POU2F3$^{high}$ (*Figure 5C*, *Figure 5—figure supplement 1C*). Because UMAP does not require the number of clusters as an a priori input (as K-means clustering does), the emergence of these five clusters in this four-dimensional space is an independent validation of the robustness of these signatures based on the (relative) levels of ASCL1, NEUROD1, YAP1, and POU2F3. Put together, these results indicate that measuring the levels of these four molecules can be sufficient to quantify the degree of phenotypic heterogeneity, thereby corroborating our current status of understanding of functional relevance of these four key molecules in SCLC.

## Discussion

The focus on existence and dynamics of non-genetic, i.e., phenotypic, heterogeneity in cancer is a relatively recent one (*Inde and Dixon, 2018*; *Jolly et al., 2018*), given a reluctantly growing realization that extensive efforts and resources to map the genomic landscape solely has had limited success in decoding any fundamental organizing principles of cancer progression, especially metastasis (*Brock and Huang, 2017*). While the existence of phenotypic heterogeneity in cancer cells is, by no means, a surprise, given its ubiquity in embryonic development (*Huang, 2009*), the extent of our lack of understanding of organizing principles enabling it is appalling, relative to our current understanding of causes and consequences of phenotypic heterogeneity in 'simpler' biological organisms (*Ackermann, 2015*; *van Boxtel et al., 2017*; *Varahan et al., 2019*).

Here, we demonstrate that the phenotypic heterogeneity in SCLC can emerge from multistable dynamics of underlying regulatory network. Multistability in this network has been reported based on Boolean (discrete, parameter-independent) simulations earlier (*Tripathi et al., 2020b*; *Udyavar et al., 2017*). Here, RACIPE – a continuous, parameter-agnostic approach – revealed that this multistability can be observed over many parametric combinations and reveal the predominance of the four states as seen in Boolean modeling, thereby elucidating the crucial role of network topology in facilitating multistability. These observations corroborate earlier work on comparative analysis of phenotypic (steady-state) distributions obtained from RACIPE and Boolean modeling for networks underlying epithelial–mesenchymal plasticity (EMP) (*Hari et al., 2020*).

The four phenotypes in SCLC reported here – ASCL1$^{high}$/NEUROD1$^{low}$, ASCL1$^{low}$/NEUROD1$^{high}$, ASCL1$^{high}$/NEUROD1$^{low}$, and ASCL1$^{high}$/NEUROD1$^{high}$ – have been seen previously using varied

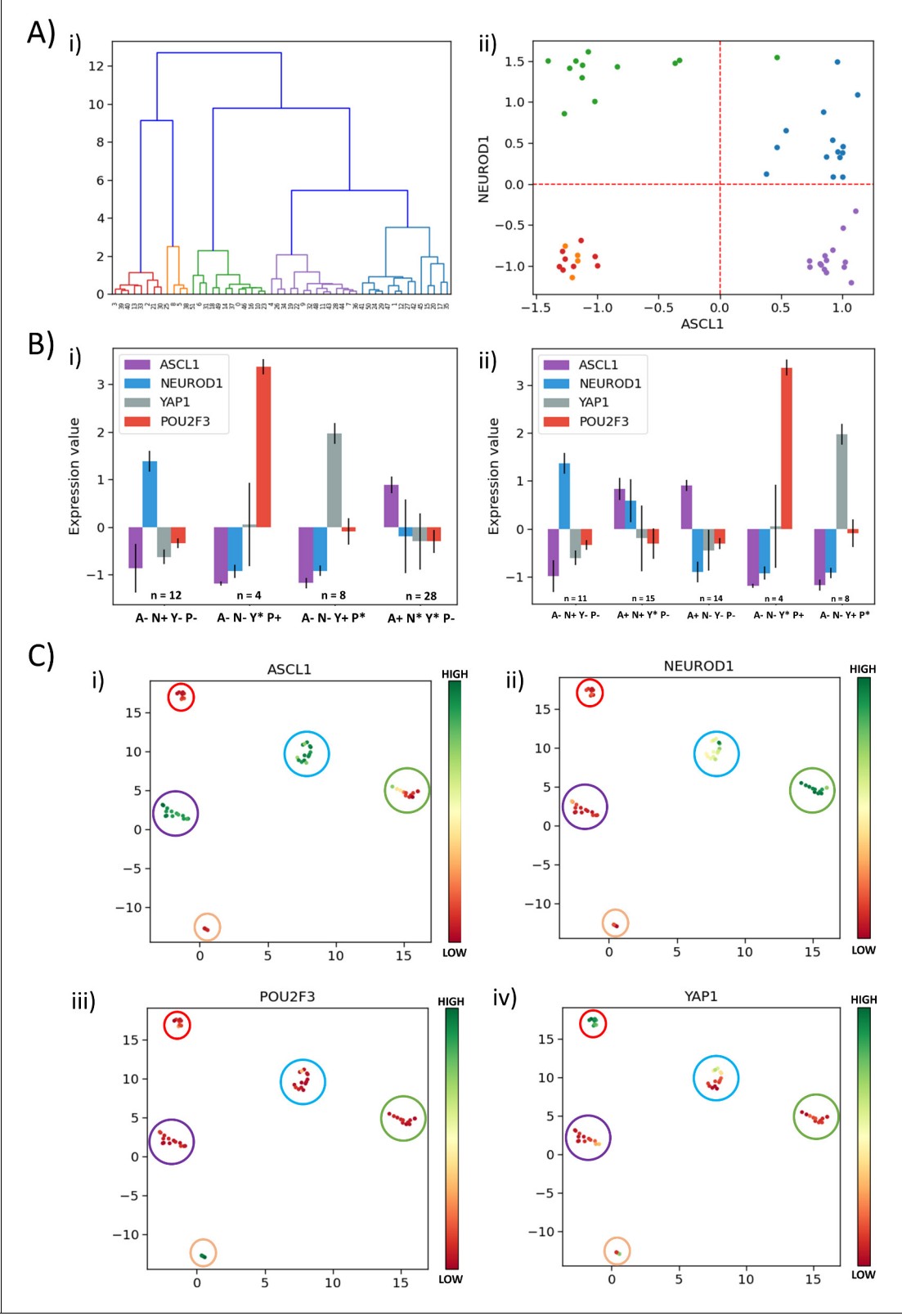

**Figure 5.** Classification of SCLC phenotypes based on ASCL1, NEUROD1, YAP1, and POU2F3. (**A**) (i) Hierarchical clustering of CCLE samples using ASCL1, NEUROD1, YAP1, and POU2F3. (ii) Scatter plot of normalized gene expression data for CCLE samples for NEUROD1 and ASCL1. Color coding in dendrogram and scatter plot are synonymous (i.e. refer to the same cluster). (**B**) (i) Average levels of expression values for ASCL1, NEUROD1, YAP1, and POU2F3 for the four clusters identified in CCLE SCLC cell lines for K = 4. (ii) Same as (i) but for K = 5. (**C**) UMAP projections for CCLE dataset from

*Figure 5 continued on next page*

*Figure 5 continued*

four dimensions (ASCL1, NEUROD1, YAP1, and POU2F3) to two dimensions. Colorbar shows individual expression levels of ASCL1, NEUROD1, POU2F3, and YAP1. Color of the enclosing circle follows the same scheme as that for dendrogram and scatter plot.

The online version of this article includes the following figure supplement(s) for figure 5:

**Figure supplement 1.** Analysis for GSE73160 using ASCL1, NEUROD1, POU2F3, and YAP1.

Boolean modeling strategies and different networks (*Tripathi et al., 2020b*; *Udyavar et al., 2017*; *Wooten et al., 2019*). Our focus is not to compare comprehensive gene expression profiles that have been mapped on to these phenotypes or to use statistical methods on transcriptomic data to infer modules of co-expressing genes, as has been extensively done before for SCLC. Rather, we use a complementary approach – RACIPE – to ascertain whether the phenotypes obtained via Boolean approaches are seen in ODE models also, and consequently identify design principles of the complex SCLC network, which constrain the number of steady states obtained. While RACIPE identifies many more steady states, this feature can be attributed to its continuous mode of simulation, a difference we also noted earlier in EMP networks (*Hari et al., 2020*). The biological significance, if any, of these additional states needs further investigation. Furthermore, both Boolean and RACIPE approaches are implemented here only on transcriptional networks, whereas additional mechanisms such as epigenetics may alter frequency of steady states by effectively altering network topology (*Somarelli et al., 2016*).

Moreover, here, we quantified the role of network topology by defining an influence matrix that revealed that the SCLC network consists of two 'teams' of players that mutually inhibit each other and self-activate themselves. This topological signature helps understand why despite such size, density, and complexity of the network, it led to a limited number of steady states. Interestingly, this property of the network was largely intact upon most single-edge perturbations, indicating enough redundancy and robustness in underlying network design. We found this property of the network is unique to this topology: not only was it lost upon randomization, but also none of the steady states obtained from random networks matched with that of the WT network topology. Importantly, we observed striking similarity between the influence matrix and the correlation matrix that included pairwise correlations among all gene expression values, endorsing that the influence matrix can provide an approximate readout of the patterns of steady states that can be expected in a network and that can be deciphered without having to simulate the dynamics of the network. However, the scalability or universality of influence matrix remains to be investigated cautiously.

Importantly, the patterns of steady states seen in influence and correlation matrices matched well with those seen in experimental data. This observation evoked confidence in biological insights that can be gained from the simulations of SCLC network. First, ASCL1 and NEUROD1, the two master regulators extensively reported in SCLC (*Jiang et al., 2009*; *Osborne et al., 2013*; *Poirier et al., 2013*), did not lie in the same 'team', consistent with recent reports that they regulate distinct set of genes and that ASCL1$^{high}$ and NEUROD1$^{high}$ cell lines can have quite distinct chromatin landscapes too (*Borromeo et al., 2016*). Second, clustering based results raised the possibility that some SCLC cell lines may represent a ASCL1$^{high}$/NEUROD1$^{high}$ state, which can be perceived as an 'intermediate' state between the canonical SCLC-A (ASCL1$^{high}$/NEUROD1$^{low}$) and SCLC-N (ASCL1$^{low}$/NEUROD1$^{high}$) ones. Similar 'intermediate' states are witnessed in diverse biological contexts (*Duddu et al., 2020*; *Jolly et al., 2015*; *Tripathi et al., 2020c*; *Yu et al., 2017*). Profiling of circulating tumor cell-derived explant models (CDXs) supports such possibility in SCLC (*Simpson et al., 2020*). Third, multistability often leads to phenotypic switching and co-existence of multiple states (heterogeneity). Mouse models of SCLC reveal phenotypically distinct cells with a NE or non-NE marker profile that cooperated to enhance their metastatic potential (*Calbo et al., 2011*), offering selective advantage to the tumor. Recent single-cell RNA-seq measurements of CDXs and patient-derived circulating tumor cells demonstrated intra-tumor heterogeneity (ITH). Particularly, ITH was found to be higher in drug-resistant CDXs relative to drug-sensitive ones and was found to have a transcriptional rather than a genomic basis (*Stewart et al., 2020*). Thus, it is expected that most SCLC tumors contain cells in these phenotypes with varying frequencies, i.e., one SCLC tumor can contain multiple subtypes within it (*Wooten et al., 2019*), as has been postulated for breast cancer (*Yeo and Guan*,

*2017*). Their relative frequency can be altered by adaptations at genetic and/or non-genetic levels (*Mollaoglu et al., 2017*; *Udyavar et al., 2017*).

A limitation of our simulations is the inability to account for the two 'non-NE' subtypes, SCLC-Y (YAP1[high]) and SCLC-P (POU2F3[high]), because the SCLC regulatory network simulated (*Udyavar et al., 2017*) does not contain YAP1 or POU2F3 as nodes. Thus, we performed clustering using ASCL1, NEUROD1, YAP1, and POU2F3 for SCLC cell lines and observed YAP1 and POU2F3 expression to be largely mutually exclusive within the 'double negative' ASCL1[low]/NEUROD1[low] cluster. These results are consistent with recent analysis of SCLC cell lines (*Ito et al., 2016*), CDXs (*Pearsall et al., 2020*), and patient samples (*Baine et al., 2020*). Furthermore, YAP1 negatively correlate with INSM1 (*McColl et al., 2017*), a crucial regulator of NE differentiation in SCLC (*Fujino et al., 2015*), which was found to be 'ON' in our A+N- and A+N+ states. Interestingly, A-N+ (ASCL1[low]/NEUROD1[high]) is classified as NE-variant (NEv1) (*Wooten et al., 2019*). Concomitantly, NEUROD1 has been associated with 'mesenchymal' (i.e., non-NE) markers (*Simpson et al., 2020*) and features such as migration (*Osborne et al., 2013*). Thus, in terms of an 'NE score', the phenotypes can be possibly expected to lie on a spectrum: ASCL1[high]/NEUROD1[low] ≈ ASCL1[high]/NEUROD1[high] > ASCL1[low]/NEUROD1[high] > ASCL1[low]/NEUROD1[low].

Our results highlight an important step in decoding the design principles of underlying regulatory network for phenotypic plasticity and heterogeneity in aggressive cancers such as SCLC. SCLC has no targeted therapy available till date (*Subbiah et al., 2020*), a limitation that is expected to be overcome by recent surge in high-throughput experimental data collection for SCLC and computational approaches to identify SCLC subtypes for clinical action (*Salgia et al., 2018*; *Stewart et al., 2020*; *Tlemsani et al., 2020*; *Udyavar et al., 2017*; *Wooten et al., 2019*). An abysmal, if any, correlation between mutational profiles and distinct SCLC subtypes (*George et al., 2015*), and multistability in SCLC network, argue for non-genetic causes of phenotypic heterogeneity. We have identified the topological signatures in underlying SCLC network that can give rise to multistability – two 'teams' of players with mutually inhibitory and self-activatory – and can result in a limited (n = 4) number of biologically relevant phenotypes, despite its overwhelming complexity (37 nodes, 357 edges). Such 'motifs' and consequent multistability appear to be the hallmarks of regulatory networks underlying cell differentiation and phenotypic plasticity (*Burda et al., 2011*; *Shiraishi et al., 2010*; *Zhou and Huang, 2011*). Given that network topology can hold an impressive amount of information about both the steady-state and dynamical behaviors, as demonstrated for multiple gene regulatory networks (*Alon, 2007*; *Feng et al., 2016*; *Ma et al., 2009*; *Santolini and Barabási, 2018*; *Gómez Tejeda Zañudo et al., 2017*), this work calls for an unprecedented approach to restrict the emergence of phenotypic heterogeneity in cancers – i.e., by 'attacking' (via tools such as CRISPR) the salient features of network topology.

## Materials and methods

### Experimental data
Gene expression profiles of 52 SCLC cell lines were downloaded from Broad Institute's CCLE expression data. Data for GSE73160 was downloaded from NCBI website.

### Normalization of experimental data
All data were normalized as per z-score normalization. The normalization was done across all the sample points (stable states or experimental data points) for a particular variable. The formula for calculating the z-score for jth observation for the ith variable is:

$$Z_{ij} = \frac{x_{ij} - \mu_i}{\sigma_i}$$

where μ is the mean, and σ is the standard deviation. For the purposes of the code, we used the **preprocessing.scale** function of the **sklearn** package in python.

### Correlation analysis
Pearson correlation co-efficient is a statistic that measures the linear correlation between two variables of interest. It depicts how strongly two variables are linearly related. It has a value between +1

and −1. A value of +1 is the total positive linear correlation, 0 is no linear correlation, and −1 is a total negative linear correlation. This static is applied to the experimental data to get the sense of how closely the expression levels of several nodes are linearly related. For the purposes of the code, the **scipy.stats.pearsonr** function of the **scipy** package in python was used. Along with the Pearson correlation coefficient, the function also returns **two-tail p-value**, which tells us whether two variables are statistically significant or not.

## Uniform Manifold Approximation and Projection

UMAP is a dimension reduction technique that can be used for visualization (similar to t-SNE) and is considered an improvement over previous dimension reduction techniques. UMAP was applied on z-score normalized data. For the purposes of the code, the **UMAP.fit_transform** function of the **umap** package in python was used. The *n_neighbours* parameter was set to 4 (after extensive trials over a broad range of values) and *n_epochs* set to 1000. The rest of the parameters was set to default.

## Hierarchical clustering

Hierarchical clustering was applied to the z-score normalized data. For purposes of code, the **cluster.hierarchy.linkage** function of the **scipy** package in python was used with the method set to *ward*. The dendrograms were plotted using the same, with the color grouping decided by setting the threshold values to desirable values.

## K-means clustering

K-means algorithm was applied to the z-score normalized data. For purposes of code, the **cluster. KMeans** function of the **sklearn** package in python was used with the *init* parameter set to *random*, the *n_int* set to 20, and *max_iter* to 300.

## Silhouette score analysis

Silhouette score is a way to determine the goodness of fit of a particular k-value in k-means clustering. It depicts how close the points in a cluster are to other points in the same cluster versus the other clusters. A higher score shows well-differentiated clustering. For the purposes of the code, we used **metrics.silhouette_score** function from the **sklearn** package in python, with the identified clusters being taken from **fit_predict** function applied to clusters obtained from the k-means algorithm outlined above.

## Influence matrix

The influence matrix is a formulation similar to that of the interaction matrix ($M$)), but gives a more comprehensive estimation of the effect of each on others in the network. For a given pair of nodes A and B, a path is a collection of serially connected edges originating from A and ending at B. The pathlength is then defined as the number of edges in such a path. Therefore, the interaction matrix depicts the influence of each node on the other for a pathlength of one. Consider the matrix $M^2$. The elements of the matrix can be written as:

$$M_{ij}^2 = \sum_{k=1}^{n} M_{ik} * M_{kj}$$

Which is the influence of ith node on the jth node when considering all paths of length 2. Similarly, each element of the matrix $M^l$ depicts the influence of nodes over pathlengths of $l$. We then calculate the influence matrix for a pathlength of $l_{max}$ by combining the matrices $M^l$ for $l<l_{max}$ as follows:

$$Inf_{lmax} = \frac{\Sigma_{l=1}^{l=l_{max}} \left( M^l / M_{max}^l \right)}{l_{max}}$$

$M_{max}$, obtained by setting all non-zero elements of the interaction matrix to 1, represents the magnitude of the maximum possible interaction for the given network topology and hence is used as the normalizing factor. The division ($\frac{M}{M_{max}}$) represents element-wise division and the range of the

elements of the resultant matrix is restricted between −1 and 1. The summation is divided by $l_{max}$ to again restrict the range of the elements between −1 and 1.

## Conversion from discrete to continuous formulation of correlation matrix

Experimental datasets gives us Pearson correlation coefficients which range from −1 to 1 (continuous values) while Boolean steady-state distribution (of both WT SCLC network and random networks) gives us correlation coefficients of either −1 or 1 (discrete values). Thus, we cannot formulate the J metric to be the same for all the cases. To make a fair comparison, we have converted the discrete values of the correlation coefficients to continuous values in the following way:

- The Pearson correlation matrix of the steady-state distribution resulting from the Ising model is calculated (as mentioned in **Correlation analysis** section)
- The values of the correlation matrix are then replaced with a random number drawn from a uniform distribution (using the **numpy.random.uniform** from **numpy** package in Python 3.6) ranging between α and one for positive values and −1 and -α for negative values, where α = 0.01. The choice of α stems from the fact that all the correlation coefficients are statistically significant, p<0.05. Therefore, the replacements must be statistically significant as well.
- For each Boolean network (WT SCLC network or Random network), 1000 such continuous correlation matrices are generated.
- J value is calculated for all the 1000 continuous correlation matrices obtained for a given network, and a mean of those 1000 values is taken as the J value for a given network.
- The value of J obtained for random networks (n = 1000) is plotted as a histogram.

Similar continuous formulation of correlation matrix was done for RACIPE correlation matrix as well, that is why the J value for RACIPE seen in *Figure 2B,ii* (J ~ 254) is smaller than that for RACIPE seen in *Figure 2A,ii* (J ~ 373), and similar to that for Boolean case as seen in *Figure 2B,ii* (J ~ 259).

## RACIPE and Boolean simulations

Please see details of Boolean and RACIPE simulations in Supplementary Information.

## Effective edge calculation and reduced networks

For a given Influence/Interaction matrix, an effective edge from a team of genes $T_i$ to another team $T_j$ is given by the metric E:

$$E = sgn\left(f_{Th}\left(\frac{\sum_i^{T_i}\sum_j^{T_j} M_{ij}}{\sum_i^{T_i}\sum_j^{T_j} M_{ij}^{Max}}\right)\right)$$

where $T_i$ and $T_j$ are teams of genes and $i$ and $j$ are indices of genes in Influence matrix/Interaction matrix belonging to teams $T_i$ and $T_j$. Here $M^{Max}$, obtained by setting all non-zero elements of the influence matrix /interaction matrix to 1, represents the magnitude of the maximum possible interaction for the given network topology and hence is used as the normalizing factor.

Real numbers which are close to zero are taken to zero by defining a threshold by $f_{Th}$ given as.,

$$f_{Th}(x) := \begin{cases} 1 \text{ if } abs(x) > Threshold \\ 0 \text{ if } abs(x) < Threshold \end{cases}$$

For the Reduced models, Threshold is taken as 0.05. Effective edges are reported using Signum function $sgn$ which extracts the sign of real number given as:

$$sgn(x) := \begin{cases} -1 & \text{if } x<0 \\ 0 & \text{if } x=0 \\ 1 & \text{if } x>0 \end{cases}$$

Using these edges, one can make reduced models from Influence matrix of any length and for a specific threshold.

To clearly understand how the two 'teams' of genes (group A: 22 genes – labeled from 1 to 22 in *Figure 2* and group B: 10 genes – labeled from 24 to 33 in *Figure 2*), subnetworks were generated

consisting of genes from individual teams. NEUROD1 is not a part of either group because it did not show any correlation with either team per se in correlation matrix.

Another interesting observation in steady states observed in *Figure 1B* is that state 1 and state 4 (and similarly, state 2 and state 3) differ only in the levels of NEUROD1 (i.e. degeneracy in NEUROD1). Given the observations stated above, we expected the subnetworks to yield two states, all on and all off. Interestingly when genes of group A are simulated using the Ising model asynchronous update, we obtained four states as opposed to the expected two. Analysis of the states revealed that a degeneracy analogous to that of NEUROD1 in WT network was being caused by ELF3 in group A (*Supplementary file 3a*, S7). Therefore, we decided to include ELF3 as an independent entity for the reduced model. However, steady-state frequencies of group B did not reveal any such degeneracy caused due to ELF3 (*Supplementary file 3c*, S9). By calculating the 'effective edges' metric described above, we observed two differences in the reduced model obtained from influence matrix (model 1; *Figure 3C*) and that obtained from interaction matrix (model 2; *Figure 3— figure supplement 2c*). Model 1, but not model 2, contains self-activation on ELF3 as well as an activatory link from team A to ELF3.

## Single-edge perturbation

We perturbed each edge of WT SCLC network in the following two ways:

1. Change of sign of the edge (A → B to A ⊣ B and vice versa)
2. Deletion of the edge (A → B/A ⊣ B to A ϕ B) where ϕ indicates the absence of a link. Every perturbed network is then simulated using the Ising model with asynchronous update for 220 random initial conditions. JSD is calculated for the resulting state distributions with respect to the steady-state distribution of WT SCLC network.

## Boolean framework – Ising model with asynchronous update

Ising model formalism uses discrete variables to represent the expression level of molecular species (such as micro-RNA or transcription factors etc.). Therefore, the state of the regulatory network (N-node network) can be represented by a sequence $\{s_i, s_i \in \{-1, 1\}\}$ called a 'Boolean Vector' of N binary variables where $s_i = 1$ represents high-expression level of $i$th node and $s_i = -1$ represents low expression of the node. In modeling the dynamics of network via this framework, the only knowledge required is whether each regulatory relationship between network nodes is activating or inhibitory. Regulatory interactions between the molecular species are represented by an N*N matrix called an 'Interaction Matrix (M)' where $M_{ij} = 1$ represents 'promotion' (or activation) of levels of $i$th node by $j$th node and $M_{ij} = -1$ represents 'inhibition' (or repression) of levels of $i$th node by $j$th node of the N-node network. The absence of any regulatory relationship between species $i$ and species $j$ is indicated by $M_{ij} = 0$.

At every discrete-time step, the expression level of a node $s_i$(t+1) is given as +1 if $\sum_{i=1}^{N} M_{ij} * s_j(t) > 0$ and $-1$ if $\sum_{j=1}^{N} M_{ij} * s_j(t) < 0$ and remains the same when $\sum_{j=1}^{N} M_{ij} * s_j(t) = 0$. The expression levels are updated using the asynchronous scheme in which a node from the N-node network is picked up at random at every discrete-time step and updated using the above non-linear relation. For large discrete-time dynamics, the network settles in a steady state, which means that the Boolean Vector is a fixed point of the above-given relation (i.e. will not change as time progresses).

## RAndom CIrcuit PErturbation

RACIPE is a tool that identifies robust dynamical properties of transcriptional regulatory networks (TRNs) by generating an ensemble of continuous network models with distinct kinetic parameters. For every continuous model of a TRN, RACIPE first generates a system of ordinary differential equations (ODEs). For a given node 'N' of the TRN and a set of input activating edges $A_i$ and input inhibiting edges $I_j$, the differential equation corresponding to the expression level of N is given as:

Here, $N$, $A_i$, and $I_j$ represent the expression levels of the species of the TRN. $G_N$ and $k_N$ denote the Production and Degradation rates, respectively. $A_{iN}^0$ is the threshold value of $A_i$ expression level at which the non-linearity in the dynamics of $N$ due to $A_i$ is seen. $n$ is termed as the Hill coefficient

and represents the extent of non-linearity in the regulation. $\lambda$ represents the fold change in the target node expression level upon over-expression of regulating nodes. The functions $H^{S+}$ and $H^{S-}$ are known as Shifted Hill functions (*Lu et al., 2013*) and represent the regulation of the target node by the regulatory node. Shifted Hill functions take the following form:

For the system of ODEs, RACIPE randomly samples the kinetic parameters from a pre-defined set of parameter ranges. At each parameter set, RACIPE integrates the model from multiple initial conditions and obtains steady states in the state space. For the current analysis, a sample size of 106 for parameters sets and 1000 for initial conditions was used. The parameters were sampled via a uniform distribution, and the ODE integration was carried out using the Runge–Kutta–Fehlberg method of numerical integration.

For the given TRN with 'n' nodes, the steady-state expression levels of the nodes were normalized in the following way:

For the ith node, $E_{in}$ is the normalized expression level of the node, $E_i$ is the steady-state expression level, $f_i$ is the normalization factor, $g_i$ and $k_i$ are the production and degradation of the $i$th node corresponding the current steady-state, and $\lambda_{ij}$ are the fold change in the expression of node $i$ due to node $j$. The normalized expression levels of all steady states are then converted into z-scores by scaling about their combined mean:

where $\overline{E_{in}}$ is the combined mean and $\sigma_{in}$ is the combined variance.

The z-scores can then be classified into Low (zero) and High (one) expression levels based on the sign of their values. This way we have discretized the continuous steady-state levels of the network for comparison with the frequency of Boolean steady states. The way to calculate the total frequency of each discrete state is by counting the occurrence in all the parameter sets. For parameter sets with n steady states, the count of each steady state is taken as 1/n, invoking the assumption that all the states are equally stable.

## Data and code availability

All codes are available at https://github.com/uday2607/CSB-SCLC; *Chauhan, 2021*; copy archived at swh:1:rev:eb4c869fe572bb0a98a6a7ce7a09631ad584200e.

## Acknowledgements

This work was supported by Ramanujan Fellowship awarded to MKJ by Science and Engineering Research Board (SERB), Department of Science and Technology (DST), Government of India (SB/S2/RJN-049/2018).

## Additional information

### Funding

| Funder | Grant reference number | Author |
| --- | --- | --- |
| Science and Engineering Research Board | SB/S2/RJN-049/2018 | Mohit Kumar Jolly |

The funders had no role in study design, data collection and interpretation, or the decision to submit the work for publication.

### Author contributions

Lakshya Chauhan, Uday Ram, Data curation, Formal analysis, Investigation, Methodology; Kishore Hari, Formal analysis, Methodology; Mohit Kumar Jolly, Conceptualization, Supervision, Funding acquisition

### Author ORCIDs

Lakshya Chauhan  https://orcid.org/0000-0002-5851-507X
Mohit Kumar Jolly  https://orcid.org/0000-0002-6631-2109

Decision letter and Author response
Decision letter https://doi.org/10.7554/eLife.64522.sa1
Author response https://doi.org/10.7554/eLife.64522.sa2

## Additional files

### Supplementary files

• Supplementary file 1. Frequency distributions for SCLC network. (a) Frequency distribution for asynchronous Boolean update of WT SCLC network using Ising update with $2^{20}$ and $2^{25}$ initial conditions over three replicates. (b) Steady-state frequency distribution for top 20 states of binarized RACIPE simulation of the network. (c) Node summary of single-edge perturbation of wild-type SCLC network.

• Supplementary file 2. Cell line classification using ASCL1, NEUROD1, YAP1 and POU2F3. (a) Cell line classification of CCLE dataset using different cluster values for k-means and hierarchical algorithm over four genes of interest (ASCL1, NEUROD1, YAP1, and POU2F3). Also contains the classification as given by *Wooten et al., 2019*. (b) Cell line classification of GSE73160 dataset using different cluster values for k-means and hierarchical algorithm over four genes of interest (ASCL1, NEUROD1, YAP1, and POU2F3). Also contains the classification as given by *Wooten et al., 2019* (for the cell lines included in both GSE73160 and CCLE).

• Supplementary file 3. Frequency distributions for reduced SCLC network. (a) Steady-state frequency distribution for asynchronous Boolean update of network for genes corresponding to GROUP A and ELF3. (b) Steady-state frequency distribution for asynchronous Boolean update of network for genes corresponding to GROUP A only. (c) Steady-state frequency distribution for asynchronous Boolean update of network for genes corresponding to GROUP B only. (b) Steady-state frequency distribution for asynchronous Boolean update of network for genes corresponding to GROUP A only. Steady-state frequency distribution for asynchronous Boolean update of network for genes corresponding to GROUP B only. Steady-state frequency distribution for asynchronous Boolean update of network for genes corresponding to GROUP B and ELF3.

• Transparent reporting form

### Data availability

Gene expression profiles of 52 SCLC cell lines were downloaded from Broad Institute's CCLE expression data. Data for GSE73160 was downloaded from NCBI website. All codes used to generate and analyze simulation data, and codes used to analyze gene expression data are available at : https://github.com/uday2607/CSB-SCLC copy archived at https://archive.softwareheritage.org/swh:1:rev:eb4c869fe572bb0a98a6a7ce7a09631ad584200e/.

The following previously published dataset was used:

| Author(s) | Year | Dataset title | Dataset URL | Database and Identifier |
|---|---|---|---|---|
| Polley E, Kunkel M, Evans D, Silvers T | 2016 | Exon expression for NCI small cell lung cancer cell line panel | https://www.ncbi.nlm.nih.gov/geo/query/acc.cgi?acc=GSE73160 | NCBI Gene Expression Omnibus, GSE73160 |

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
