## [Decision Letter]

**Acceptance summary:**

This paper provides a detailed analysis of the master regulatory network involved in small cell lung cancer. Using multiple mathematical tools, the authors transform a complex, highly connected network into a small, easy-to-interpret reduced form. The reduced network provides insight into the topological features – here the existence of two "teams" of genes that inhibit each other – that determine the non-genetic phenotypic plasticity in these cancer cells. The concordance between the reduced network and experimental knowledge from the field suggest that these methods may be useful more broadly to make sense of complex regulatory networks elsewhere.

**Decision letter after peer review:**

Thank you for submitting your article "Topological signatures in regulatory network enable phenotypic heterogeneity in small cell lung cancer" for consideration by *eLife*. Your article has been reviewed by three peer reviewers, and the evaluation has been overseen by a Reviewing Editor and Aleksandra Walczak as the Senior Editor. The following individuals involved in review of your submission have agreed to reveal their identity: Jean Clairambault (Reviewer #1); David Wooten (Reviewer #3).

The reviewers have discussed the reviews with one another and the Reviewing Editor has drafted this decision to help you prepare a revised submission.

The reviewers appreciated your detailed analysis of the regulatory network involved in small cell lung cancer. However, they also raise some concerns that are expressed in the reviews appended below. In your revision, please try to address the concerns of the reviewers, in particular please expand your analysis and comparison to similar results in literature (Reviewer 2) and discuss how your methods could be generalized to apply to other networks (Reviewer 3).

Reviewer #1:

Detailed study of the master regulatory network of SCLC by probabilistic methods, focusing on the two genes ASCL1 and NEUROD1. Supports by up-to-date literature reports and sound gene expression analysis the non-genetic phenotypic plasticity of SCLC cells. May be a major step forward in understanding non-genetic phenotype plasticity in cancer.

Firstly, let me say that the technical aspects of this very detailed study go far beyond my field of expertise. I will evaluate it from an external eye, and according to the contribution it brings to the investigation of non genetic phenotype plasticity in cancer, here focusing on small cell lung cancer (SCLC). Plasticity in cancer cells is indeed a major field of research nowadays, resulting in metastasis and drug persistence/tolerance, and especially SCLC has escaped efficacious treatments until now.

As far as I could understand the analysis, by two different methods, one based on Boolean networks, the other on elementary (nevertheless of great dimensionality) ordinary differential equations, the authors statistically “shake” the possible parameters of the regulatory network of genes found “masterly” expressed in SCLC, according to a study of 2017 (Udyavar et al., 2017), network in which one can recognize 3 of the 4 Yamanaka genes, MYC, KLF4 and *SOX2*.

Rather than on the Yamanaka genes, the authors focus on the two genes ASCL1 and NEUROD1, for both of which they find marked differential expression (high/low), resulting in 4 different coupled phenotypes at the gene expression level, which, as “patterns of steady states seen in influence and correlation matrices”, correlate with experimental results collected from experimental data on SCLC. Indeed, ASCL1 and NEUROD1 had already been identified as master regulators of the salient gene network of SCLC, ASCL1 being a neuroendocrine marker and NEUROD1 being associated with mesenchymal markers susceptible of inducing metastasis, i.e., two markers of plasticity.

The design of this analysis relies on an apparently complete analysis of the scientific literature on both the subject of non-genetic phenotypic heterogeneity and the particular case go gene expression studies on SCLC. The results reported by the authors support by gene expression analyses the idea that within the same genomic status of SCLC cells, different steady states in gene expression lead to different phenotypic states (multistability) that are found experimentally. Although no clear therapeutic perspective for SCLC emerges from this study, it is an interesting step to better understand the plasticity of SCLC cells, possibly resulting in their drug tolerance and thus escaping treatments so far.

For these reasons, I propose that this study be published as it is in *eLife*.

Reviewer #2:

This paper will be of interest to researchers working on the origins of cellular heterogeneity in cancer. Using network and gene expression analysis, previous known findings on the potential role of topology in generating four broad cellular states in small cell lung cancer (SCLC) are confirmed here. Interestingly, it is observed that both discrete and continuous gene expression models can give rise to similar steady states. Two modules of genes inhibiting each other are found to lead to the four emergent states in SCLC. However, these results are not well compared and contrasted in the context of prior literature.

Chauhan, et al. investigate the origins of cellular heterogeneity in Small Cell Lung Cancer. Previous experimental work has classified four major states based upon the expression levels of Ascl1 and Neurod1 genes, along with other states. Previous computational work based on Boolean networks (Wooten et al., 2019 and Udyavar et al., 2017) suggested these four states may arise naturally as the steady states of the underlying gene-interaction network in SCLC.

Here the authors re-analyze the SCLC gene-interaction network based on a Boolean framework with an underlying Ising hamiltonian. They show that the four major states based on Ascl1 and Neurod1 can be recovered from dynamic simulations of the 33 node 357 edge network. While not surprising in the light of prior work, the authors successfully demonstrate that a somewhat different Boolean framework leads to similar results, suggesting the underlying topology is indeed important for generating cellular heterogeneity. However, it is not clear whether this formalism based upon the Ising Model improves upon the previously used Boolean framework (Wooten et al. and Udyavar et al) in any way.

The authors then model the dynamics of the gene network using ODEs, allowing continuous values for the expression levels of the nodes. 4 of the top 10 frequent steady states corresponded to the 4 states found earlier. However, these 4 states have much lower frequency (adding up to only about 22%) whereas in the Boolean framework the frequencies added up to almost 100%. The identity and frequencies of the remaining 6 out of 10 frequent states is not discussed.

Using a pairwise correlation strategy for all the nodes, the authors then uncover an interesting result: the nodes fall into two modules that repress each other. This also seems to be true in two publicly available gene expression datasets. Surprisingly, Neurod1 did not appear in these modules and it is unclear why, given the four main states are defined based on Neurod1 and Ascl1 expression levels. The two modules are also distinct from the higher number of modules found earlier (Wooten et al., 2019), and it is unclear why these differences arise.

Finally, using a series of clustering algorithms (hierarchical and K-means clustering, UMAP) on gene expression datasets, the authors show that not only the 4 states based on Neurod1 and Ascl1, but also further states based on Pou2f3 and Yap1 expression levels can be recovered. However, the authors use only a few genes of interest in performing the clustering and it is not clear why all the available genes were not used.

In my opinion, the most significant area where this manuscript needs to be strengthened is in providing critical comparisons with prior literature and results (primarily Wooten et al and Udyavar et al). Discussions on what advances have been made in this paper with respect to what was already known earlier, need to be highlighted. I found it quite hard to judge this manuscript and place it in context, since a lot of the methods and datasets used here are very similar to the previous works. Detailed suggestions along these lines as well as some possible new analyses are provided below:

1) It would be good to know why the authors chose an Ising Model – based Boolean simulation strategy as compared to the Boolean model used in Wooten at al. Is there some difference in the statistics expected from these two different formalisms? Is there some limitation of the previous work that the authors wanted to address here? Given that Wooten et al. showed that the 4 SCLC states can be recapitulated, is it surprising that the authors get the same 4 states using their Boolean method on an identical network?

2) For the ODE method, the frequency of the four states add up to only about 22%. It would be interesting to see a full list of the top ten states with their frequencies, and a discussion on why these other states appear in the ODE but not in the Boolean formalism and its biological implications.

3) Following up on point (2) above, was there a reason for using two separate axes for the same quantity (frequency) in Figure 1D i ? I found this quite confusing, because for example, at first sight it seems like the S2 steady state has similar frequencies in RACIPE vs Boolean. But the frequencies are in reality very different, right? I would therefore suggest to plot both RACIPE and Boolean results using just one axis, to avoid confusion.

4) The observation of two "modules" using pair-wise correlations is interesting. However, it was unclear to me why Wooten et al. find 17-18 modules, though their WGCNA method also uses a pair-wise gene correlation technique. A detailed discussion on this would be very helpful for readers in my opinion.

5) Related to the pair-wise correlation method, I was surprised to see that Neurod1 does not seem to be part of any module in Figure 2. In the Discussion, the authors mention that Ascl1 and Neurod1 don't fall in the same team, but it seems to me from Figure 2 that Neurod1 doesn't belong to any team! This seems to be contradictory to the rest of the results, unless I have misunderstood something here. A discussion on these lines seems warranted.

6) Given that the dynamical simulations were carried out with 33 genes, why did the authors choose to perform all the clustering analyses with only a handful of genes? This may be problematic, for example, if sets of 2 or 4 randomly chosen genes are used for clustering the expression datasets, how likely are we to find a few well separated clusters? If we find that random gene subsets also separate into clusters, how biologically meaningful is it to see clusters with Ascl1 and Neurod1?

Reviewer #3:

This work by Chauhan et al. finds order from complexity. Using multiple mathematical tools, they transform a complex, highly connected network into a small, easy-to-interpret reduced form. The concordance between their reduced network and experimental knowledge from the field support the insights they gained from their analysis here, and that their methods may be useful more broadly to make sense of complex networks elsewhere.

The final findings (Figures 4-5) on experimental data, which show that just a few genes can reproduce the full-spectrum of known SCLC heterogeneity seem to strongly support the authors' primary conclusions. Specifically, a system which is well classifiable using a few nodes closely matches my expectation for a network that can be reduced to a few tightly connected groups of nodes.

The agreement between Boolean and RACIPE greatly strengthen their results.

The J metric is interesting, and while it would be beyond the scope of this work, I would be very interested in seeing it applied to other networks to see how generalizable its interpretation is. However, the definition is tightly coupled to the 33 genes in this network, and their pattern of expression in the steady states. Discussion about how the metric could generalized would strengthen the manuscript.

I found it very interesting that not only Neurod1, but also Elf3 was pulled out as an individual node in the reduced form. In another work (Wooten et al., 2019), the authors identified ELF3 as a master regulator of one of their subtypes (NEv2). The identification of this node through topological features here also lends further credence to the influence matrix.

The authors state that "These results suggest that influence matrix is a better representation of network topology as compared to the interaction matrix.". However, since the influence matrix comes from the interaction matrix, it seems like it necessarily contains less information. The authors make this claim based on the fact that a network reduction based on influence matrix more closely represents the steady state distributions than a similar reduction based on interaction matrix. But it is not clear how much this conclusion is specific to this particular network, or reduction strategy.

The correspondence of the steady states with expression data appears quite promising! However, the fact that Neurod1 is the sole gene that distinguishes S1 from S4, or S2 from S3, makes me suspect other genes must also contribute to the difference? Are there other genes in the literature that the authors think could be included into new versions the network that could give a broader picture of the differences between S1 vs S4, or S2 vs S3? Given the other 31 nodes in the network, do their steady state values more closely match one or another cluster from Figure 4B?

When introducing the Font-Clos s_i_(t+1) equation, I recommend to describe what happens if s_i_=0, rather than just including that info in supplement.

Figure 1B should have a legend indicating dark=off, blank=on (even though it is in the caption)

I do not see what test / method was used to find the +/- % confidence intervals in Figure 1B, nor what size interval they represent (e.g., 95%?)

The reference in-text to Figure 1C, i, regarding swapping random edges, seems to actually refer to both i and ii

In the text, the connection between the larger number of steady states of "random" networks to the true network's topology lacks a relevant reference to Figure 1C, iii

The text introducing the J metric should describe what the indices are, rather than requiring the reader to search the figure.

The introduction of influence matrix was very hard to follow, the grammar is confusing, and "lmax" is not clearly described in the main text, even though it is used several times.

[Editors' note: further revisions were suggested prior to acceptance, as described below.]

Thank you for resubmitting your work entitled "Topological signatures in regulatory network enable phenotypic heterogeneity in small cell lung cancer" for further consideration by *eLife*. Your revised article has been evaluated by Aleksandra Walczak (Senior Editor) and a Reviewing Editor.

The reviewers are happy with the revisions you have made and the paper is almost ready for acceptance, but one substantive point raised by Reviewer 2 remains (see below). Please try to perform clustering analysis for randomly chosen sets of 2 genes (other than ASCL1 and NEUROD1). If these don't result in well-separate clusters it strengthens your result, but if they do then you can modify your statements accordingly, for instance by expanding on other biological reasons to focus on ASLC1 and NEUROD1 based clustering. Alternatively, if you feel such clustering based on randomly chosen genes is not useful, please provide some reasons why you think so.

Reviewer #2:

The authors have now satisfactorily responded to most of the comments/queries.

Their response to point (6) that I had raised is not entirely satisfactory however, since they do not seem to have addressed the randomization point that I had raised. If random sets of two (or more) genes are chosen for the clustering analysis, how often do we see well separated clusters? It seems to me an important point to analyze and understand, in order to put the ASCL1 and Neurod1 based clustering in perspective. I would strongly urge the authors to include this analysis, unless they feel this is not a sensible question to address, in which case it would be good to hear their arguments against this.

Reviewer #3:

The revised manuscript addresses my original comments, and I support its publication.

---

## [Author Response]

Reviewer #2:In my opinion, the most significant area where this manuscript needs to be strengthened is in providing critical comparisons with prior literature and results (primarily Wooten at al and Udyavar et al). Discussions on what advances have been made in this paper with respect to what was already known earlier, need to be highlighted. I found it quite hard to judge this manuscript and place it in context, since a lot of the methods and datasets used here are very similar to the previous works. Detailed suggestions along these lines as well as some possible new analyses are provided below:1) It would be good to know why the authors chose an Ising Model – based Boolean simulation strategy as compared to the Boolean model used in Wooten at al. Is there some difference in the statistics expected from these two different formalisms? Is there some limitation of the previous work that the authors wanted to address here? Given that Wooten et al. showed that the 4 SCLC states can be recapitulated, is it surprising that the authors get the same 4 states using their Boolean method on an identical network?

We thank the reviewer for this important question. Udyavar et al., 2017 – the precursor manuscript to Wooten et al., 2019 – had also used the Ising model-based Boolean simulation strategy. Thus, as far as Figure 1B is concerned, these are for the same circuit (Figure 1A) and modeling strategy as in Udyavar et al.

Wooten et al. have used a different network than what was used in Udyavar et al. and also a different modeling strategy (BooleaBayes) which depends on inferring logical relationships between nodes in gene regulatory networks using gene expression data and using a “Bayes-like adjustment approach”. Therefore, the network Wooten et al. arrived at using BooleaBayes (Figure 7A) is different from the one we and Udyavar et al. have used. Hence, obtaining 4 states in this network may not necessarily be surprising as far as SCLC phenotypes are concerned. However, obtaining only four states from such a complex network (33 nodes, 357 edges) was very intriguing to us, which drove further our investigations in network topology-based analysis.

The key message in our manuscript is not that this network has only four states, but the reasons for why only four states are seen in this complex and large network (33 nodes, 357 edges). We have deciphered a “latent” design principle in SCLC network, and have offered a conceptual framework to decode similar design principles hidden in other regulatory networks.

Moreover, in addition to running Ising model based simulations for the network, we have used a parameter-agnostic approach – RACIPE – and notice at least a semi-quantitative agreement in terms of dominance of the four states identified via Ising model. This agreement strengthens our key point that network topology alone can contain enough information about its dynamics. We have now added a paragraph in the Discussion section highlighting these salient points.

2) For the ODE method, the frequency of the four states add up to only about 22%. It would be interesting to see a full list of the top ten states with their frequencies, and a discussion on why these other states appear in the ODE but not in the Boolean formalism and its biological implications.

We thank the reviewer for this subtle observation and raising this important question. We have now performed a detailed analysis of top 20 states obtained by RACIPE whose frequencies add up to 54%, and observed that these 20 states are very similar to the top 4 states obtained by Boolean (X1-X4 in Figure 1B), with a difference noted in only one or two nodes (i.e. node value = 0 in Boolean state and 1 in RACIPE state or *vice versa*).

It is not surprising to see that the RACIPE output has a much larger number of states than Boolean output, given its continuous nature, an observation we made in our previous manuscript as well (Hari et al., 2020) and is now included in a new paragraph in the Discussion.

Conceptually speaking, given that the sizes of the two “groups” identified are 22 and 10 nodes, a difference in values of one or two nodes are not very likely to give rise to a completely different biological phenotype. Thus, these “close-enough” states can be thought of as “micro-states” that overall constitute a biological “macro-state” or phenotype.

3) Following up on point (2) above, was there a reason for using two separate axes for the same quantity (frequency) in Figure 1D i ? I found this quite confusing, because for example, at first sight it seems like the S2 steady state has similar frequencies in RACIPE vs Boolean. But the frequencies are in reality very different, right? I would therefore suggest to plot both RACIPE and Boolean results using just one axis, to avoid confusion.

Thanks to the additional analysis performed (mentioned in response to point (2)); we have now replaced Figure 1D i to address this point.

4) The observation of two "modules" using pair-wise correlations is interesting. However, it was unclear to me why Wooten et al. find 17-18 modules, though their WGCNA method also uses a pair-wise gene correlation technique. A detailed discussion on this would be very helpful for readers in my opinion.

The WGCNA method gives us correlation data from all the genes whose expression values are used as an input to the algorithm. It is a statistical method that works on threshold based correlation, and does not use any mechanistic information embedded in a network topology. Because network topology information is not required, Wooten et al. were able to start with a much larger set of genes and obtain a large number of modules. Not surprisingly, we found that the 33 genes considered here were spread across different modules. This is not surprising or contradictory because any of these 33 genes can still be correlated strongly with any other gene not in the network simulated here, and those genes may belong to different modules. It should be noted that two genes can show a good correlation in their gene expression values without any of them directly or indirectly affecting each other, say transcription factor activates gene B but inhibits gene C, thus, B and C are most likely to be negatively correlated. In brief, our analysis is based on network topology, not transcriptomic data that is input to WCGNA. This point is now included in a new paragraph added in the Discussion section.

5) Related to the pair-wise correlation method, I was surprised to see that Neurod1 does not seem to be part of any module in Figure 2. In the Discussion, the authors mention that Ascl1 and Neurod1 don't fall in the same team, but it seems to me from Figure 2 that Neurod1 doesn't belong to any team! This seems to be contradictory to the rest of the results, unless I have misunderstood something here. A discussion on these lines seems warranted.

We apologize for a potential semantic confusion caused. The two statements – “NEUROD1 does not belong in the same team as ASCL1” and “NEUROD1 does not belong to either of the two teams” – are not contradictory to one another. We have performed further analysis based on CCLE and GSE73160 gene expression values and see that NEUROD1 levels do not align with the patterns seen in the expression values of members of two groups (Figure 4—figure supplement 2).

6) Given that the dynamical simulations were carried out with 33 genes, why did the authors choose to perform all the clustering analyses with only a handful of genes? This may be problematic, for example, if sets of 2 or 4 randomly chosen genes are used for clustering the expression datasets, how likely are we to find a few well separated clusters? If we find that random gene subsets also separate into clusters, how biologically meaningful is it to see clusters with Ascl1 and Neurod1?

Our reason to include ASCL1 and NEUROD1 based clustering was purely based on available experimental literature suggesting these two nodes as key markers and/or inducers of phenotypic heterogeneity in SCLC. In the latter half of our analysis (Figure 5), we have used YAP1 and POU2F3 in addition to ASCL1 and NEUROD1 to classify CCLE SCLC cell lines, but we do not have either of them as a part of the network in the first place. Again, their choice was made based on available literature in SCLC heterogeneity as mentioned by Wooten et al.

Reviewer #3:[…]The authors state that "These results suggest that influence matrix is a better representation of network topology as compared to the interaction matrix.". However, since the influence matrix comes from the interaction matrix, it seems like it necessarily contains less information. The authors make this claim based on the fact that a network reduction based on influence matrix more closely represents the steady state distributions than a similar reduction based on interaction matrix. But it is not clear how much this conclusion is specific to this particular network, or reduction strategy.

We are grateful to the reviewer for encouraging remarks and we share the excitement about agreement in Boolean and RACIPE results, as well as a broader application of influence matrix. We have already seen preliminary evidence showing the two “teams” regulating phenotypic heterogeneity in other cancer-related signaling networks (for instance, see Figure 2 in Jia et al., 2020 – https://www.oncotarget.com/article/27651/text/). Because these additional networks are not directly related to SCLC, we are not including those results in this manuscript, but influence matrix based analysis will be the focus of our upcoming manuscript(s).

The correspondence of the steady states with expression data appears quite promising! However, the fact that Neurod1 is the sole gene that distinguishes S1 from S4, or S2 from S3, makes me suspect other genes must also contribute to the difference? Are there other genes in the literature that the authors think could be included into new versions the network that could give a broader picture of the differences between S1 vs S4, or S2 vs S3? Given the other 31 nodes in the network, do their steady state values more closely match one or another cluster from Figure 4B?

We have included additional analysis (Figure 4—figure supplement 2) highlighting the expression of other nodes besides ASCL1 and NEUROD1. As expected, the two groups identified via influence matrix largely show similar trends, i.e. as compared to A-N- and A-N+ subgroups, members of group A (ASCL1 is one of them) have higher expression levels in A+N- and A+N+ samples while members of group B have lower expression levels in these two subgroups. This analysis suggests that while other genes could play a role in defining the subtypes, their contribution might not possess the distinguishing power of ASCL1 and NEUROD1.

When introducing the Font-Clos s_i_(t+1) equation, I recommend to describe what happens if s_i_=0, rather than just including that info in supplement.

We have now included the same.

Figure 1B should have a legend indicating dark=off, blank=on (even though it is in the caption)

We have included this legend in Figure 1B.

I do not see what test / method was used to find the +/- % confidence intervals in Figure 1B, nor what size interval they represent (e.g., 95%?)

We have now clarified it in the figure legend. The +/- % represent mean and standard deviation of the frequencies obtained across three independent Ising-model replicates.

The reference in-text to Figure 1C, i, regarding swapping random edges, seems to actually refer to both i and ii

We have now referred to both i and ii.

In the text, the connection between the larger number of steady states of "random" networks to the true network's topology lacks a relevant reference to Figure 1C, iii

We thank the reviewer for pointing this out and have now included a reference to Figure 1C, iii.

The text introducing the J metric should describe what the indices are, rather than requiring the reader to search the figure.

In the main text, we have now described what the indices and the corresponding matrix is.

The introduction of influence matrix was very hard to follow, the grammar is confusing, and "lmax" is not clearly described in the main text, even though it is used several times.

We apologize for the confusion caused, and thank the reviewer for pointing it out. We have now expanded on the introduction of influence matrix both in the Materials and methods section and in the main text.

[Editors' note: further revisions were suggested prior to acceptance, as described below.]

Reviewer #2:The authors have now satisfactorily responded to most of the comments/queries.Their response to point (6) that I had raised is not entirely satisfactory however, since they do not seem to have addressed the randomization point that I had raised. If random sets of two (or more) genes are chosen for the clustering analysis, how often do we see well separated clusters? It seems to me an important point to analyze and understand, in order to put the ASCL1 and Neurod1 based clustering in perspective. I would strongly urge the authors to include this analysis, unless they feel this is not a sensible question to address, in which case it would be good to hear their arguments against this.

We thank the reviewer for this constructive comment, and have performed the randomization analysis as well now. We find that the ASCL1-NEUROD1 gene pair is among the top 1% of all possible gene pairs (^33^C_2_ = 528) in terms of defining four SCLC phenotypes experimentally reported (Figure 4—figure supplement 3). In other words, approximately 99% of all gene pairs considered here do not offer such well-segregated biologically relevant phenotypic distinction.